# (How) Do Language Models Track State?

**Belinda Z. Li** [1]   **Zifan Carl Guo** [1]   **Jacob Andreas** [1]

## Abstract

Transformer language models (LMs) exhibit behaviors—from storytelling to code generation—that seem to require tracking the unobserved state of an evolving world. How do they do this? We study state tracking in LMs trained or fine-tuned to compose permutations (i.e., to compute the order of a set of objects after a sequence of swaps). Despite the simple algebraic structure of this problem, many other tasks (e.g., simulation of finite automata and evaluation of boolean expressions) can be reduced to permutation composition, making it a natural model for state tracking in general. We show that LMs consistently learn one of two state tracking mechanisms for this task. The first closely resembles the "associative scan" construction used in recent theoretical work by Liu et al. (2023) and Merrill et al. (2024). The second uses an easy-to-compute feature (permutation parity) to partially prune the space of outputs, and then refines this with an associative scan. LMs that learn the former algorithm tend to generalize better and converge faster, and we show how to steer LMs toward one or the other with intermediate training tasks that encourage or suppress the heuristics. Our results demonstrate that transformer LMs, whether pre-trained or fine-tuned, can learn to implement efficient and interpretable state-tracking mechanisms, and the emergence of these mechanisms can be predicted and controlled. Code and data are available at `https://github.com/belindal/state-tracking`.

## 1. Introduction

Language models (LMs) are trained to model the surface form of text. A growing body of work suggests that model internals contain a latent, decodable state of the world—e.g., situations described by language and results of program execution—to support prediction (Li et al., 2021; Nanda et al., 2023; Li et al., 2023). However, the mechanisms that LMs use to construct these representations are not understood. Do LMs simulate state evolution step by step across successive hidden layers or token representations (Yang et al., 2024)? Are states approximated through a complex collection of heuristics (jylin04 et al., 2024)? Is state tracking an illusion (Bender & Koller, 2020)?

This paper studies the implementation and emergence of state tracking mechanisms in language models using permutation composition as a model system: given a fixed set of objects, we train or fine-tune LMs to predict the final position of each object after a sequence of rearrangements. Previous work has used versions of this task to evaluate LMs' empirical state tracking abilities (Li et al., 2021; Kim & Schuster, 2023; Li et al., 2023). Additionally, as shown by Barrington (1989), many complex, natural, state-tracking tasks—including simulation of finite automata and evaluation of Boolean expressions—can be reduced to permutation tracking with five or more objects. This makes it a natural model for studying state tracking in general.

Our analysis proceeds in several steps. §2 provides technical preliminaries: §2.1 and §2.2 introduce state tracking problems and the permutation composition task we use to model them (Figure 1A), and §2.3 reviews the set of interpretability tools we use to analyze LM computations. Next, §3 lays out a family of algorithms that past work has suggested LMs might, in principle, use to solve the state tracking task (Figure 1D), and describes the **signatures**—expected readouts from different interpretability methods—that we would expect to find if a given algorithm is implemented (Figure 1B-C).

Finally, §4 and §5 present experimental findings. Across a range of sizes, architectures, and pretraining schemes, we find that LMs consistently learn one of two state tracking mechanisms. The first mechanism, which we call the "associative algorithm" (AA), resembles the associative scan construction used by Liu et al. (2023) and Merrill & Sabharwal (2024) to establish theoretical lower bounds on the expressive capacity of Transformers. The second mechanism, which we call the "parity-associative algorithm" (PAA), first rules out a subset of final states using an easy-to-compute permutation parity heuristic, then uses an associative scan

---

[1]MIT EECS and CSAIL. Correspondence to: Belinda Z. Li <bzl@mit.edu>.

*Proceedings of the 42nd International Conference on Machine Learning*, Vancouver, Canada. PMLR 267, 2025. Copyright 2025 by the author(s).

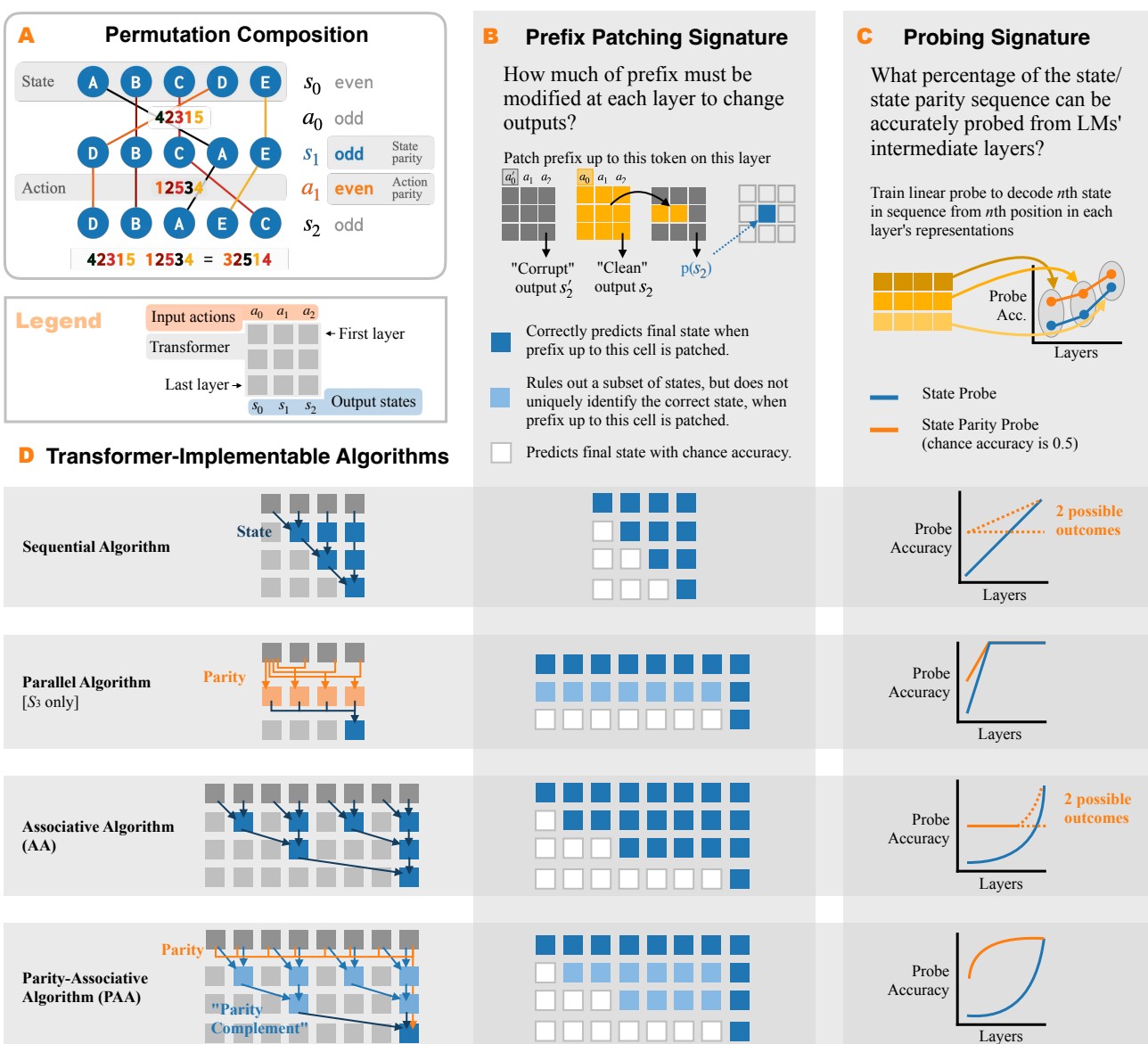

*Figure 1.* We use permutation word problems as a simple model of state tracking. Actions are permutations, and states are the products of those permutations; the current state can be tracked by taking the cumulative product from left to right (§2). We identify several possible algorithms that Transformers may use to solve permutation word problems: sequential, parallel, associative, and parity-associative (§3). Above, we depict the "signatures" of each algorithm under two types of interpretability analysis: *prefix patching*, where we create pairs of prompts differing only on the first token, then substitute all activation except the prefix up to a token at a particular layer, and *probing*, where we train a linear probe to map from last-token representations across the layers to either the final state or the final state parity (§2.3). Note: the dotted lines indicate two different probing signatures consistent with this algorithm (see Appendix C.3 for more details).

to obtain a final prediction. Notably, we fail to find evidence for either step-by-step simulation or fully parallel composition, despite their being theoretically implementable by LMs. We support our findings with evidence from representation interventions (Meng et al., 2022; Zhang & Nanda, 2024; §4.2), probes (Shi et al., 2016; §4.3), patterns in prediction errors (Zhong et al., 2024; §4.4), attention maps (Clark et al., 2019; §4.5), and training dynamics (McCoy et al., 2020;

Olsson et al., 2022; Hu et al., 2023; §5.1).

The scan operation for PAA appears difficult for LMs to implement robustly, and the choice of mechanism sometimes significantly impacts model performance on long sequences (§5.1). Whether a given LM learns AA or PAA is highly stochastic (§5.2). However, each is associated with a characteristic set of phase transitions in the training loss (Chen et al., 2024), and LMs can be steered toward one solution or

the other by training on an intermediate task that encourages or discourages LMs from learning a parity heuristic (§5.3).

As pretrained LMs sometimes re-use circuits when fine-tuned on related tasks (Prakash et al., 2024; Merullo et al., 2024), our results suggest a possible mechanism by which real-world LMs might perform state tracking when modeling language, code, and games. We show preliminary evidence of these algorithms on a version of our permutation composition tasks expressed in natural language (Appendix E). Looking beyond state tracking, these findings underscore both the complexity and variability of LM solutions to complex tasks, which may involve *both* heuristic features and structured solutions.

## 2. Background and Preliminaries

### 2.1. State Tracking

Inferring common ground in discourse (Li et al., 2021), navigating the environment (Vafa et al., 2024), reasoning about code (Merrill et al., 2024), and playing games (Li et al., 2023; Karvonen, 2024) all require being able to track the evolving state of a real or abstract world. There has been significant interest in understanding whether (and how) LMs can perform these tasks. In theoretical work, researchers have observed that many natural state-tracking problems (including the ones listed above) are associated with the complexity class $NC^1$, but Transformers cannot track the state of arbitrarily long inputs (Merrill et al., 2022; Huet et al., 2025; Bhattamishra et al., 2020; Delétang et al., 2023; Strobl et al., 2024). However, prior work has shown that Transformers with $O(\log n)$ depth can model inputs of up to length $n$ (Liu et al., 2023; Merrill & Sabharwal, 2024). Empirical work, meanwhile, has found that large LMs learn to solve state tracking problems (Kim & Schuster, 2023) and encode state information in their representations (Li et al., 2021; Li et al., 2023). But a mechanistic understanding of *how* trained LMs infer these states has remained elusive.

### 2.2. Permutation Group Word Problems

Toward this understanding, the experiments in this paper focus on one specific state tracking problem, *permutation composition*. At a high level, this problem presents LMs with a set of objects and a sequence of reshuffling operations; LMs must then compute the final order of the objects after all reshufflings have been applied (Figure 1A). Though less familiar than discourse tracking or program evaluation, Kim & Schuster (2023) used a version of this task to evaluate LM state tracking. More importantly, as shown by Barrington (1989) and recently highlighted by Merrill et al. (2024), permutation tracking (with five or more objects) is $NC^1$-complete, meaning *any* other state tracking task in this family can be converted into a permutation tracking class.

This, combined with its simple structure, makes it a natural model system for studying state tracking in general.

More formally, the **finite symmetric group** $S_n$ comprises the set of **permutations** of $n$ objects equipped with a composition operation. For example, 42315 denotes the permutation of 5 objects (i.e. in $S_5$) that moves the first object to the fourth position, the second object to the second position, etc. Importantly for our findings in this paper, every permutation can be expressed as a composition of two-element **swaps** (in Figure 1A, $a_0$, but not $a_1$, is an example of a swap). The **parity** of a permutation (even or odd) is the parity of the number of swaps needed to create it.

The **composition** of two permutations, standardly denoted $a_1 \circ a_0$, is the result of applying $a_1$ after $a_0$. Inputs to sequence models in machine learning are typically written with earlier inputs before later inputs (i.e., left-to-right), so for consistency with this convention, we will write $a_0 a_1$ to denote the application of $a_0$ *then* $a_1$. Figure 1A shows the result of composing 42315 and 12534 in sequence.

Finally, the **word problem on** $S_n$ is the problem of computing the product of a sequence of permutations. This product itself corresponds to a single permutation (32514 in Figure 1). But, following the intuition given at the beginning of the section, it may equivalently be interpreted as the final ordering of the objects being rearranged (DBAEC in Figure 1A). Following this intuitive explanation (and by analogy to other state tracking problems), we will use $a_t$ to denote a single permutation ("action") in a sequence, and $s_t = a_0 \cdots a_t$ to denote the result of a sequence of permutations (a "state").

Given a sequence of permutations, we use $\epsilon(a_t)$ to denote the parity of the $t$th permutation, so:

$$\epsilon(s_t) = \epsilon(a_0 \cdots a_t) = \sum_i \epsilon(a_i) \mod 2 \qquad (1)$$

(where $\epsilon$ is 0 for even permutations and 1 for odd ones).

All experiments in this paper train transformer language models to solve the word problem: they take as input a sequence of actions $[a_0, \ldots, a_t]$, and output a sequence of state predictions $[s_0, \ldots, s_t]$. We also validate our findings on a natural language version of this task in Appendix E, where permutations are expressed as instructions like *swap positions 2 and 3*.

### 2.3. Interpretability Methods

Our experiments employ several interpretability techniques to understand how LMs solve permutation word problems, which we briefly describe below. Throughout this paper, we use $h_{t,l}$ to denote the internal LM representation at token position $t$ after Transformer layer $l$, with $T$ and $L$ denoting the maximum input length and number of layers respectively.

**Probing**   In probing experiments (Shi et al., 2016), we fix the target LM, then train a smaller "probe" model (e.g. a linear classifier) to map LM hidden representations $h$ to quantities $z$ hypothesized to be encoded by the LM (Figure 1C). Our experiments specifically evaluate whether (1) the state $s_t$, and (2) the final state parity is linearly encoded in intermediate-layer representations. For each layer $l$, we train (1) a state probe to predict $p(s_t \mid h_{t,l})$ and (2) a parity probe to predict $p(\epsilon(s_t) \mid h_{t,l})$. Given a trained LM, we collect representations on one set of input sequences to train the probe, then evaluate probe accuracy on a held-out set.

**Activation Patching**   Probing experiments reveal what information is *present* in an LM's representations, but not that this information is *used* by the LM during prediction. Activation patching is a method for determining which representations play a *causal* role in prediction. Portions of the LM's internal representations are overwritten ("patched") with representations derived from alternative inputs; if predictions change, we may conclude that the overwritten representations was used for prediction (Meng et al., 2022; Zhang & Nanda, 2024; Heimersheim & Nanda, 2024).

Let $p(y \mid x; h \leftarrow h')$ denote the probability that an LM assigns to the output $y$ given an input $x$, but with the representation $h$ replaced by some other representation $h'$. In a typical experiment, we first construct a "clean" input $x$ that we wish to analyze, and a "corrupted" input $x'$ that alters or removes information from $x$ (e.g. by adding noise or changing its semantics). Next, we compute the most probable outputs from clean and corrupted inputs:

$$\widehat{y} = \arg \max_y p(y \mid x)$$
$$\widehat{y'} = \arg \max_y p(y \mid x')$$

We then re-run the LM on the corrupted input $x'$, but substitute a hidden representation from the clean input $x$, and measure how much prediction shifts toward the clean output $\hat{y}$ using the *normalized logit difference* (Wang et al., 2023):

$$\text{NLD} = \frac{\text{LD}(x'; h_{t,l} \leftarrow h_{t,l}^{\text{clean}}) - \text{LD}(x')}{\text{LD}(x) - \text{LD}(x')} \qquad (2)$$

where

$$\text{LD}(\cdot) = \log p(\widehat{y} \mid \cdot) - \log p(\widehat{y'} \mid \cdot)$$

and the representation $h_{t,l}^{\text{clean}}$ is taken from the clean run of the model. A value of NLD close to 1 indicates that we have *restored* a part of the circuit that computes $\hat{y}$.

In this paper, we evaluate which representations are involved in prediction by presenting models with a clean sequence $[a_0, a_1, \ldots, a_t]$ associated with a final state $s_t$. We then produce a corrupted sequence differing **only in the first token**, $[a'_0, a_1, \ldots, a_t]$, associated with a final state

$s'_t$. We then identify the hidden states that, when patched in, cause the model to output $s_t$ rather than $s'_t$ with high probability. Our main experiments specifically perform *prefix patching*, where *all* hidden representations up to index $t$ ($h_{1:t,l} \leftarrow h_{1:t,l}^{\text{clean}}$) are patched at a particular layer $l$ (Figure 1B). Prefix patching allows us to localize how information gets progressively transferred to the final token as we move deeper into the network. A value close to 1 means that some part of the prefix representation was used for prediction; a value close to 0 means that *no* part was.

We also experiment with other types of localization techniques (including suffix and window patching), as well as zero-ablating certain activations in Appendix B.

## 3. What Algorithms Can Transformers Implement in Theory?

To use the methods described in §2.3 to interpret model behavior, we must first establish a *phenomenology* for LM state tracking—identifying candidate state tracking algorithms that might be implemented by the model, along with the empirical probing and activation patching results we would expect to find if these algorithms are implemented. Below, we describe a set of state tracking mechanisms suggested by the existing literature.

For each mechanism, we first present a sketch of an implementation, in the form of rules for computing the value stored in the hidden state for each layer and timestep. We then describe the "signature" of each algorithm—the result we would expect from the application of prefix patching and probing techniques described in the preceding section.

### 3.1. Sequential Algorithm

The sequential algorithm composes permutations one at a time from left to right (analogous to a mechanism some LMs use to solve multi-hop reasoning problems; Yang et al., 2024). Signatures of this algorithm would provide evidence that LMs implement step-by-step "simulation" in their hidden states to solve state tracking tasks. In this algorithm, each hidden state $h_{t,l}$ stores the associated action $a_t$ until $s_t$ can be computed, maintaining $h_{t,t} = s_t$. As shown in the first row of Figure 1D, this computation depends only on hidden states with $l \leq t$.

$h_{t,0} = a_t \ \forall t$      // initialize actions
$(h_{0,0} = s_t)$      // by definition; see §2.2
**for** $t = 1..T, l = 1..L$ **do**
    **if** $l < t$ **then** $h_{t,l} = h_{t,l-1} = a_t$    // propagate actions
    **if** $l = t$ **then** $h_{t,l} = h_{t-1,l-1} h_{t,l-1}$
                   $= s_{t-1} a_t = s_t$    // update states
    **if** $l > t$ **then** $h_{t,l} = h_{t,l-1} = s_t$    // propagate states
**end for**

**Patching Signature** Because of this dependency, any patching experiment that replaces only hidden states with $l > t$ will not affect the final model predictions, leading to the upper triangular patching signature shown in the first row of Figure 1B.

**Probing Signature** Because $s_t$ can only be predicted at layer $l = t$, we expect a state probe to show a *linear* dependence on depth: for sequences of maximum length $T$, a probe at layer $l$ will correctly label an $l/T$ fraction of states. If these state representations linearly encode parity, then the accuracy of the parity probe will also increase linearly; otherwise, it will remain constant.[1]

### 3.2. Parallel Algorithm

As noted in §2.2, the word problem on $S_5$ belongs to $\mathsf{NC}^1$ (and thus requires a circuit depth that scales logarithmically with sequence length). The word problem on $S_3$, however, belongs to $\mathsf{TC}^0$, the class of decision problems with constant-depth threshold circuits. See discussion in Appendix A and Merrill & Sabharwal (2023). A constant-depth circuit will give rise to a set of hidden-state dependencies like the second row of Figure 1D.

**Patching Signature** Let $l_P$ denote the number of layers needed to implement the constant-depth circuit for this task. For patching interventions conducted at or earlier than layer $l_P$, we expect the model's predictions to change; at deeper layers than $l_P$, interventions will have no effect at all, resulting in the L-shaped pattern shown in the second row of Figure 1B.

**Probing Signature** We expect the probe to obtain perfect accuracy within a constant number of layers. Because the algorithm described in Appendix A computes state parity as an intermediate quantity, the parity probe will also obtain perfect accuracy within a constant number of layers.

### 3.3. Associative Algorithm

In the associative algorithm (AA), Transformers compose permutations hierarchically: in each layer, adjacent sequences of permutations are grouped together and their product is computed. This is analogous to recursive scan in Liu et al. (2023) and flattened expression evaluation in Merrill et al. (2024). This algorithm takes advantage of the associative nature of the product of permutations, whereby $a_0 a_1 a_2 a_3 = (a_0 a_1)(a_2 a_3)$. It ensures that $h_{t,l} = a_{t-2^l+1} \cdots a_t$, and thus that $h_{t,\log(t+1)} = a_0 \cdots a_t$. Signatures of this algorithm would provide evidence that LMs perform state tracking not by encoding states, but rather by *mapping* between states, for prefixes of increasing length.

---

[1]See Appendix C.3 for a representative model in which parity is not linearly decodable.

$$
\begin{aligned}
&h_{t,0} = a_t \;\; \forall t && \text{// initialize actions} \\
&\textbf{for } t = 0..T, l = 1..L \textbf{ do} \\
&\quad \textbf{if } l \le \log(t+1) \textbf{ then} \\
&\quad\quad h_{t,l} = h_{t-2^{l-1},l-1} h_{t,l-1} \\
&\quad\quad\quad = a_{t-2^l+1} \cdots a_t && \text{// compose actions} \\
&\quad \textbf{else } h_{t,l} = h_{t,l-1} = s_t && \text{// propagate actions} \\
&\textbf{end for}
\end{aligned}
$$

(Defining $h_{t<0,l} = h_{0,l}$ for notational convenience.)

As seen in the third row of Figure 1D, the model's prediction for $s_t$ depends on the hidden representation $h_{t/2}$ in the layer before the final state is computed, the representation at $h_{t/4}$ in the layer before that, etc.

**Patching Signature** Consequently, for AA, the length of the prefix that must be modified to alter model behavior increases exponentially in depth, resulting in the signature in the third row of Figure 1B.

**Probing Signature** We similarly expect to see an exponentially increasing state probe accuracy (because $s_t$ becomes predictable at layer $l = \log t$, a probe at layer $l$ will correctly label a $2^l/T$ fraction of states). If state parity is encoded in state representations, then parity probe accuracy will also increase exponentially.

### 3.4. Parity-Associative Algorithm

In this algorithm (PAA), LMs compute the final state in two stages: first computing the parity of the state (which can be performed in a constant number of layers using a subroutine from the Parallel algorithm); then separately computing the remaining information needed to identify the final state (the "parity complement") using a procedure analogous to AA. (Unlike the preceding algorithms, we are not aware of any previous proposals for solving permutation composition problems in this way; but as we will see, it is useful for understanding interactions between "heuristic" and "algorithmic" solutions in real LMs.)

We model implementation of PAA with hidden states comprising two "registers" $\epsilon$ and $\kappa$ (i.e. $h_{t,l} = (\epsilon_{t,l}, \kappa_{t,l})$ which store the parity and complement respectively.

$$
\begin{aligned}
&\kappa_{0,t} = a_t \;\; \forall t && \text{// initialize actions} \\
&\epsilon_{0,t} = \mathsf{par}(s_t) \;\; \forall t && \text{// compute parities (App. A)} \\
&\textbf{for } t = 0..T, l = 1..L \textbf{ do} \\
&\quad \epsilon_{t,l} = \epsilon_{t,l-1} && \text{// propagate parities} \\
&\quad \textbf{if } l \le \log(t+1) \textbf{ then} \\
&\quad\quad \kappa_{t,l} = \mathsf{comp}(\kappa_{t-2^{l-1},l-1}\kappa_{t,l-1}) && \text{// compose} \\
&\quad \textbf{else } \kappa_{t,l} = \kappa_{t,l-1} && \text{// propagate complements} \\
&\textbf{end for}
\end{aligned}
$$

In this algorithm, the hidden state at position $i$ holds that

position's state parity and parity complement (if computed at this point). Parity, like $S_3$, may be computed with a constant number of layers. The algorithm sketch given above is deliberately vague about the implementation of the parity complement composition operation (comp). In practice, different representations of this complement appear to be learned across different runs; see Figure 10 for evidence that these representations are computed using a brittle (and perhaps heuristic- or memorization-based) mechanism.

**Patching Signature**    If the corrupted input has a different parity from the clean input, then in layers deeper than those used to compute parity, it is necessary to restore the *entire* prefix to cause the LM to assign full probability to the clean prediction. On these inputs, prefix patching will show a signature similar to the parallel algorithm (see Figure 8B). However, if the corrupted input has the *same* parity as the clean input, the portion of the hidden state computed in parallel remains the same, while its complement is computed using the same mechanism as the associative algorithm (see Figure 8A). These inputs will thus exhibit an AA-like (exponentially-shaped) patching pattern. When averaged together, parity-matched and parity-mismatched patching will produce a pattern with two regions, one shaped like the associative algorithm (associated with a 50% restoration in accuracy) and one shaped like the parallel algorithm (associated with a 100% restoration in accuracy). Again, this may be most easily understood graphically (Figure 1).

**Probing Signature**    We expect state probes to improve exponentially with depth, while parity probes converge to 100% at a constant depth.

# 4. What Mechanisms do Transformers Learn?

In this section, we compare these theoretical state tracking mechanisms to empirical properties of LMs trained for permutation tasks. It is important to emphasize that the various signatures described above provide necessary, but not sufficient, conditions for implementation of the associated algorithm; the exact *mechanism* that LMs use in practice is likely complex and dependent on other input features not captured by the algorithms described above.

Nevertheless, our experiments successfully rule out some possible state tracking mechanisms and identify algorithmic features likely to be shared between the idealized mechanisms above and the true behavior learned by transformers. Specifically, our experiments yield evidence consistent with the associative algorithm (AA) in some models and the parity-associative algorithm (PAA) in other models, across architectures, sizes, and initializations.

## 4.1. Experimental Setup

We generate 1 million unique length-100 sequences of permutations in both $S_3$ and $S_5$. We split the data 90/10 for training/analysis, and fine-tune these models (using a cross-entropy loss) to predict the state corresponding to each *prefix* of each action sequence:

$$\mathcal{L} = -\sum_{t=0}^{99} \log p_{\mathsf{LM}}(s_t \mid a_0 \dots a_t), \qquad (3)$$

where $p_{\mathsf{LM}}(s_n \mid a_0 \dots a_t)$ is the probability the language model places on state token $s_n$ when conditioned on the length-$n$ prefix of the document.

Except where noted, we begin with Pythia-160M models pre-trained on the Pile dataset (Biderman et al., 2023). Regardless of initialization scheme, we fine-tune models for 20 epochs on Equation (3) using the AdamW optimizer with learning rate 5e-5 and batch size 128. For larger models (above 700M parameters), we train using bfloat16.

## 4.2. Activation Patching

For both the $S_3$ and $S_5$ tasks, across training runs, we find that activation patching results exhibit two broad clusters of behavior. For some trained models, they match the activation patching signature associated with AA; in others, they match the signature of PAA—even when the only source of variability across training runs is the order in which data is presented. Results for prototypical AA- and PAA-type models, on both $S_3$ and $S_5$, are shown in Figure 2. Additional patching results in Appendix B confirm that patching intermediate representations of PAA-type models (the light-

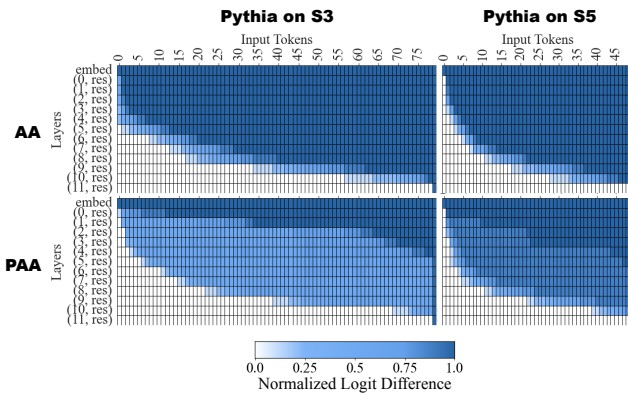

*Figure 2.* Activation patching on the residual stream for various Pythia models trained on $S_3$ and $S_5$. Each cell at layer $l$ and token $t$ represents the probability of the correct final state when *the entire prefix up to $t$ at layer $l$* is restored. We find signatures matching the AA and PAA algorithms from Figure 1, with both models ignoring exponentially longer prefixes as we traverse down the layers, and PAA models containing intermediate representations that encode some information about the final state, but not its parity.

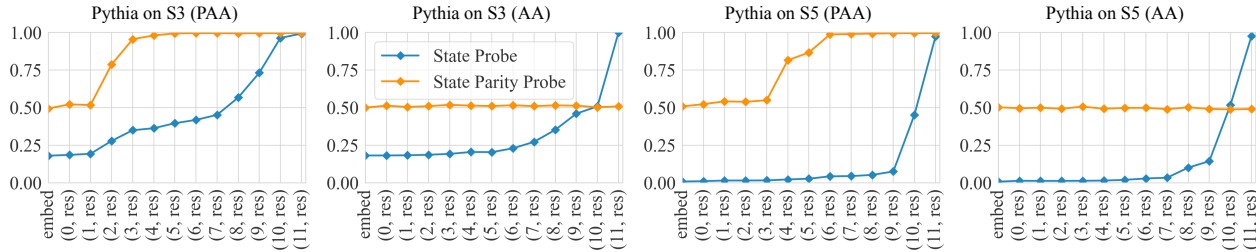

*Figure 3.* Accuracy of state probe and state parity probe across layers on $S_3$ and $S_5$ models sometimes match signatures for AA, and sometimes PAA. In all models, the state probe accuracy increases roughly exponentially with model depth. We find that in PAA models, the parity of the state is linearly decodable from earlier intermediate layers, while in the AA models shown above, the parity is never linearly encoded in any layer of the model. (In other AA models, the parity can only be linearly decoded at the final layer.)

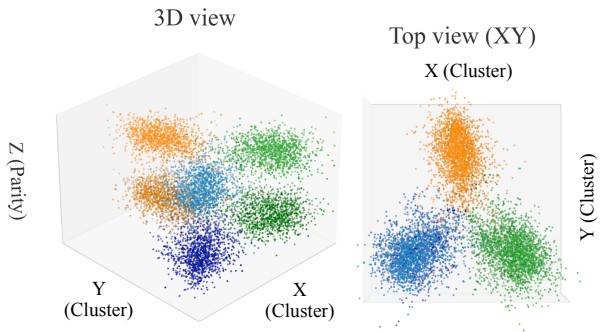

*Figure 4.* In models that learn PAA on $S_3$, representations of the final product can be geometrically decomposed into two orthogonal directions, corresponding to the *parity* of the product (represented as the Z-axis in the above graph) and *cluster identity* of the product (represented by the X-Y plane). Note that the *clusters* are at 60 degrees to each other, and products of different parities within a cluster are equidistant from each other, with odd-parity products in one plane, and even-parity products in another plane.

colored cells in Figure 2) results specifically in predictions with incorrect parity. We find standard deviations to be low in Figure 9, confirming the robustness of our signatures.

### 4.3. Probing

Test set accuracies of linear probes across LM layers $l$ are plotted in Figure 3. We report standard deviations of these accuracies in Table 1, which are all less than $10^{-3}$. We again find empirical signatures consistent with those predicted by AA and PAA, on both $S_3$ and $S_5$. Models with AA-type probing signatures always have AA-type patching signatures, and vice versa. Throughout the rest of this paper, we refer to models (and state-tracking mechanisms) as "AA-type" or "PAA-type" based on which cluster of signatures they exhibit. Results in Appendix C break down probe accuracies by sequence length, confirming that models solve sequences of exponentially longer length at deeper layers.

What exactly is the "non-parity residual" for PAA models? We visualize the linear components of representations near the final layer(s) of PAA models trained on $S_3$. The rep-

resentations of states can be cleanly decomposed into two orthogonal parts: the *parity* of the product and a residual *cluster identity*, forming a triangular prism. In Figure 4, we project representations from the PAA model for each of the six states onto these components. Even-parity states (darker colors) and odd-parity states (lighter colors) are symmetric. The three cluster "spokes" are spaced 60 degrees apart.[2]

### 4.4. Generalization by Sequence Length

We next evaluate the state and parity accuracy of AA- and PAA-type models for held-out inputs of varying length. In general, we find that models learn to generalize perfectly to sequences of up to the length of their training data, then face a steep accuracy dropoff after (which we refer to as the "*cutoff length*"), rather than generalizing uniformly across all sequence lengths.

In Figure 5, where we plot the cutoff lengths at which each accuracy dips below 98%. We find that for models that learn PAA, the *parity accuracy cutoff length* is much longer than the *state accuracy cutoff length*, whereas, for models that learn an AA-type mechanism, the two cutoff lengths are equal. Furthermore, models that learn an AA-type mechanism tend to generalize better overall.

### 4.5. Attention Patterns

We look at attention patterns of LMs and check whether they can be used to differentiate between PAA models and AA models. Specifically, we find that in the early layers, PAA models exhibit *parity heads*, heads that place attention to *odd-parity* actions. Recall that the parity of a state can be determined by counting the number of odd-parity actions, and taking the parity of the count (Equation (1)). Examples of the parity head attention pattern are shown in Figure 15. We find no evidence of parity heads in any layer of AA models. (See Appendix F for a formal metric measuring how much an attention head behaves like a parity head.)

---

[2]Further details, including an analysis of $S_5$, can be found in Appendix D.

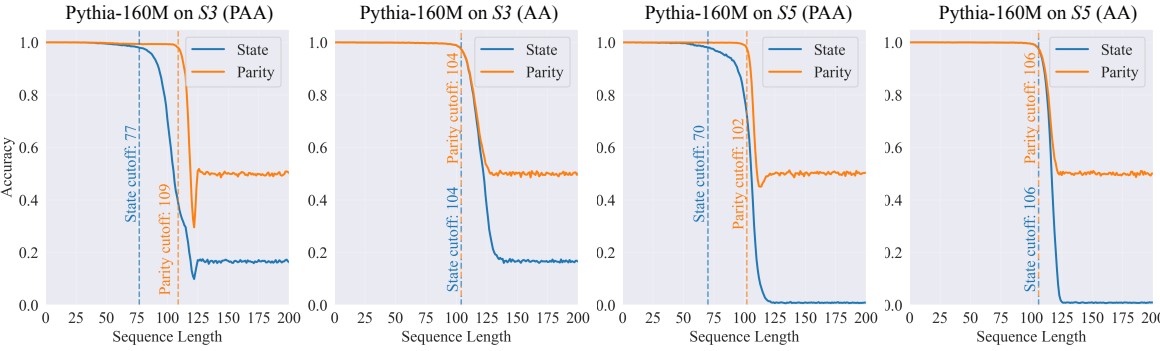

*Figure 5.* Generalization curves showing state and parity prediction accuracy as sequence lengths vary. Models are trained on length-100 sequences and asked to generalize to varying lengths of sequences. We plot generalization curves for AA and PAA models on $S_3$ and $S_5$. In each plot, we show the 98% cutoff threshold, the sequence length at which accuracy dips below 98%. In the models that learned PAA, the parity cutoff is larger than the state cutoff, while in models that learned AA, the parity cutoff equals the state cutoff. Generally speaking, models that learned AA generalize better than ones that learned PAA.

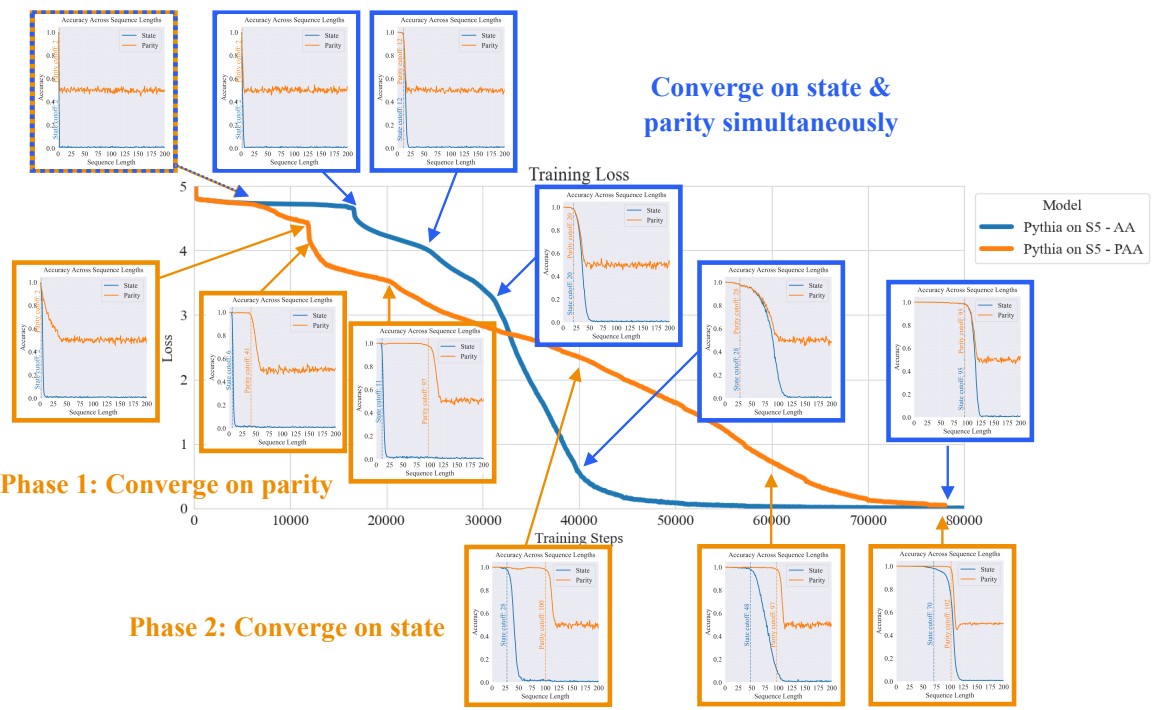

*Figure 6.* Annotated training curves for models that learn the AA and PAA algorithms. In PAA models (blue), we find that convergence happens in two phases: in the first phase, they learn to generalize parities up to sequence length 100, and in the second, they learn to generalize the *states*. In AA models (orange), parities and states are learned simultaneously. Note that AA models also tend to converge faster to (ultimately) a lower loss than PAA models.

We also find evidence that attention patterns in AA models sparsify in later layers of the network, forming a tree-like pattern expected of AA, shown in Figure 16.

# 5. Why do Transformers Learn One Mechanism or Another?

Having determined that trained models consistently exhibit AA- or PAA-like signatures, we next study the factors that determine *which* mechanism emerges during training.

## 5.1. When in Training Do Distinct Mechanisms Arise?

We find that an LM's eventual mechanism can be identified very early in training, based on the pattern of prediction errors. As in Figure 6, LMs that eventually learn AA improve the quality of their parity and state predictions in lockstep, while LMs that learn PAA learn in two phases: they first converge on learning *parity* over the entire length of the

training sequence; and only then do they learn to accurately predict the *state itself*.[3]

Because it is possible to identify these patterns early in training, our subsequent experiments classify LMs as AA-type or PAA-type based on generalization curves (Section 4.4) after 10k training steps, rather than waiting for the full probing and patching signatures to emerge.

### 5.2. What Factors Affect Which Mechanism is Learned?

Whether an LM learns AA or PAA is a deterministic function of four factors: model architecture, size, initialization scheme, and fine-tuning data order. Our next experiments evaluate each of these factors in turn. We explore two different model architecture families of various sizes (GPT-2, Radford et al., 2019, and Pythia, Biderman et al., 2023), several different model initializations (pre-trained on the Pile and trained from scratch with different random initializations), and up to 12 different data ordering seeds.

We find that model architecture and initialization, rather than model size, are the biggest determining factors of what mechanism the model chooses to learn. Figure 7 shows the ratio of LMs that learn each mechanism, aggregated by model architecture and initialization. The low variance indicates a minimal effect of model size.[4] GPT-2 models, pre-trained or not, are split roughly evenly between the two mechanisms, while Pythia models tend to learn AA when pre-trained on the Pile, and PAA when not.

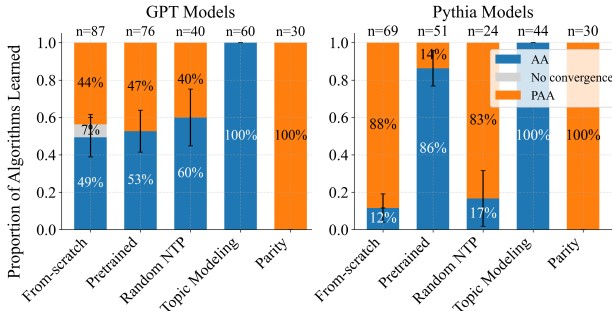

*Figure 7.* Proportion of GPT-2 and Pythia models that learn an AA-type mechanism, a PAA-type mechanism, or neither under different training regimes described in Section 5.

### 5.3. How Does Pre-training Affect Mechanism Choice?

We show that appropriately designed intermediate tasks can encourage models to learn one mechanism or the other.

**Topic Modeling**  As a controlled way of studying how the next-token-prediction (NTP) objective affects which

mechanism LMs converge to, we generate length-100 documents with only $S_3$ elements as vocabulary items, and pre-train (randomly initialized) LMs with NTP on these documents, before training them on $S_3$. Specifically, the documents are generated from a topic model with parameters: # of topics $= 4$, $\alpha = 0.3$, and $\beta = 0.1$, where $\alpha$ is the density of topics in each document and $\beta$ is density of words in each topic. As shown in Figure 7, when from-scratch LMs are trained with our topic modeling NTP objective, they always learn an AA-type mechanism.

**Parity Prediction**  We first train the entire model on predicting the **state parity** of the sequence to output token 1 if odd and 0 if even, before transitioning to training on the actual $S_3$ objective. In Figure 7, we show that we can induce GPT-2 and Pythia models to learn PAA when trained from scratch on parity. Notably, from-scratch Pythia models already tend to learn PAA as a baseline behavior. Therefore, we also apply this curriculum on Pythia models pre-trained on the Pile, and find that it consistently converts the mechanism learned from AA-type to PAA-type.[5]

**Control: Random Next-Token-Prediction**  As a control, we train LMs on length-100 documents of *random $S_3$* elements sampled from a uniform distribution. We confirm that the control fine-tuning did not affect the ratio with which LMs learned each mechanism.

## 6. Conclusion

We have shown that LMs trained on permutation tracking tasks learn one of two distinct mechanisms: one consistent with an "associative algorithm" (AA) that composes action subsequences in parallel across successive layers; and another with a "parity-associative algorithm" (PAA) which first computes a shallow parity heuristic in early layers and then computes a residual to the parity using an associative procedure. LMs that learn an AA-type mechanism tend to generalize better and converge faster; different choices of model architecture and training scheme encourage the discovery of one mechanism over another.

While a large number of other state tracking tasks can be reduced to the more complex permutation task we study ($S_5$), our experiments leave open the question of whether the specific mechanisms LMs use to solve $S_5$ are also deployed for these other tasks.

## Impact Statement

The $S_3$ and $S_5$ tasks we choose to study in this paper can be generalized to many different state tracking scenarios funda-

---

[3]Experimental details can be found in Appendix G.

[4]A finer-grained breakdown of the ratio over each model size can be found in Figure 17.

[5]We discuss another parity curriculum using an extra parity loss term in Appendix H.3.

mental to many aspects of reasoning capabilities. Methods for identifying mechanisms that LMs implement, especially when these differ from human-designed algorithms, can provide crucial insights on how to build more robust LMs, control their behavior, and predict their failures. Our experiments focus on small-scale models, and we do not anticipate any immediate ethical considerations associated with our findings.

## Acknowledgments

This work was supported by the OpenPhilanthropy foundation, the MIT Quest for Intelligence, and the National Science Foundation under grant IIS-2238240. BZL is additionally supported by a Clare Boothe Luce fellowship, and JA is supported by a Sloan fellowship. This work benefited from many conversations during the Simons Institute Program on Language Models and Transformers. The authors would also like to thank Reuben Stern, Sebastian Zhu, and Gabe Grand for feedback on drafts of the paper.

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

# A. A Constant-Depth Algorithm Exists for $S_3$

$S_5$ is the smallest non-solvable permutation group. $S_3$ is isomorphic to $D_3$, the symmetry group of an equilateral triangle, which can be generated by a transposition $a = (23)$ and a 3-cycle $b = (123)$.[6] These generators satisfy the relation $ab = ba^{-1}$, which allows any word problem in $S_3$ to be reduced by tracking (1) the cumulative parity of transpositions and (2) the count of 3-cycles modulo 3. Since both parity checking and modular counting can be computed using constant-depth threshold circuits, the word problem for $S_3$ belongs to $\mathsf{TC}^0$.

# B. Full Activation Patching Results

In the activation patching experiments, we overwrite ("patch") portions of the LM's internal representations and compute how much the resulting logits have changed. As discussed in Section 2.3, we perform prefix patching to localize important token positions. However, in addition to prefix patching, we also explore the following types of localization methods:

1. In *suffix patching*, all tokens starting from $t$ up to *one before* the last token of the sequence ($h_{t:|A|-1,l}$) are patched at a particular layer $l$.[7]

2. In *window patching*, all tokens in a $w$-width window starting from $t$ ($h_{t:t+w-1,l}$) are patched at layer $l$.

For each of the above localization techniques, we patch the representation with several different *types of content*:

1. In *representation deletion*, we overwrite target representation(s) entirely with a zero vector,

$$h_{t,l} = \mathbf{0}$$

and measure the NLD as follows:

$$\text{NLD} = \frac{\text{LD}(a_1 \ldots a_t) - \text{LD}(a_1 \ldots a_t; h_{t,l} \leftarrow \mathbf{0})}{\text{LD}(a_1 \ldots a_t)}$$

2. In *representation substitution*, we overwrite the representation(s) with those derived from running the LM on a minimally different (corrupted) representation $P_{\text{corrupt}}$. This is the setting described in Section 2.3.

Full results are shown in Figure 8. In general, we discover the following:

**In PAA models, parities are computed in parallel in early-mid layers**  We use prefix substitution patching described in Section 2.3, but plot pairs that have *same parity* final states ($\epsilon(\widehat{y}) = \epsilon(\widehat{y'})$) separately from pairs that have *opposte parity* final states ($\epsilon(\widehat{y}) \neq \epsilon(\widehat{y'})$).

Results are shown in Figure 8A (for same parity) and 8B (for opposite parity). We find that in AA models, parities are computed with the state – with both the same-parity and opposite-parity patching patterns displaying the same exponential curve. However, in PAA models, the patching patterns for same- and opposite-parities differ drastically. When parities are the same, only the parity complement must be computed to infer the final state; the patching pattern in this case indicates that the parity complement is computed in an associative manner. When the parities are different, the patching signature has a component that resembles a parallel patching signature, which is where the parity is computed. Restoring prefixes of layers before the parity is computed results in the entire prediction being of the correct parity, while restoring prefixes of layers after that results in the entire prediction being of the incorrect parity. We see that parities are computed roughly in parallel at early layers (around layers 3-5). Note that there is a middle region where restoring the prefixes shifts the logits towards the correct prediction, but not 100%: when prefixes in these regions are restored, the LM does not know the parity of the final answer, but does know some aspects of the parity complement, which was computed in an associative manner.

---

[6] https://proofwiki.org/wiki/Symmetric_Group_on_3_Letters_is_Isomorphic_to_Dihedral_Group_D3

[7] We do not patch the last token as the last token residual contains the current accumulated information necessary for computing the final product, and almost always will destroy the prediction when patched, making this method uninformative as localization tool. We wish to see what *other* tokens the final token uses when constructing its representation.

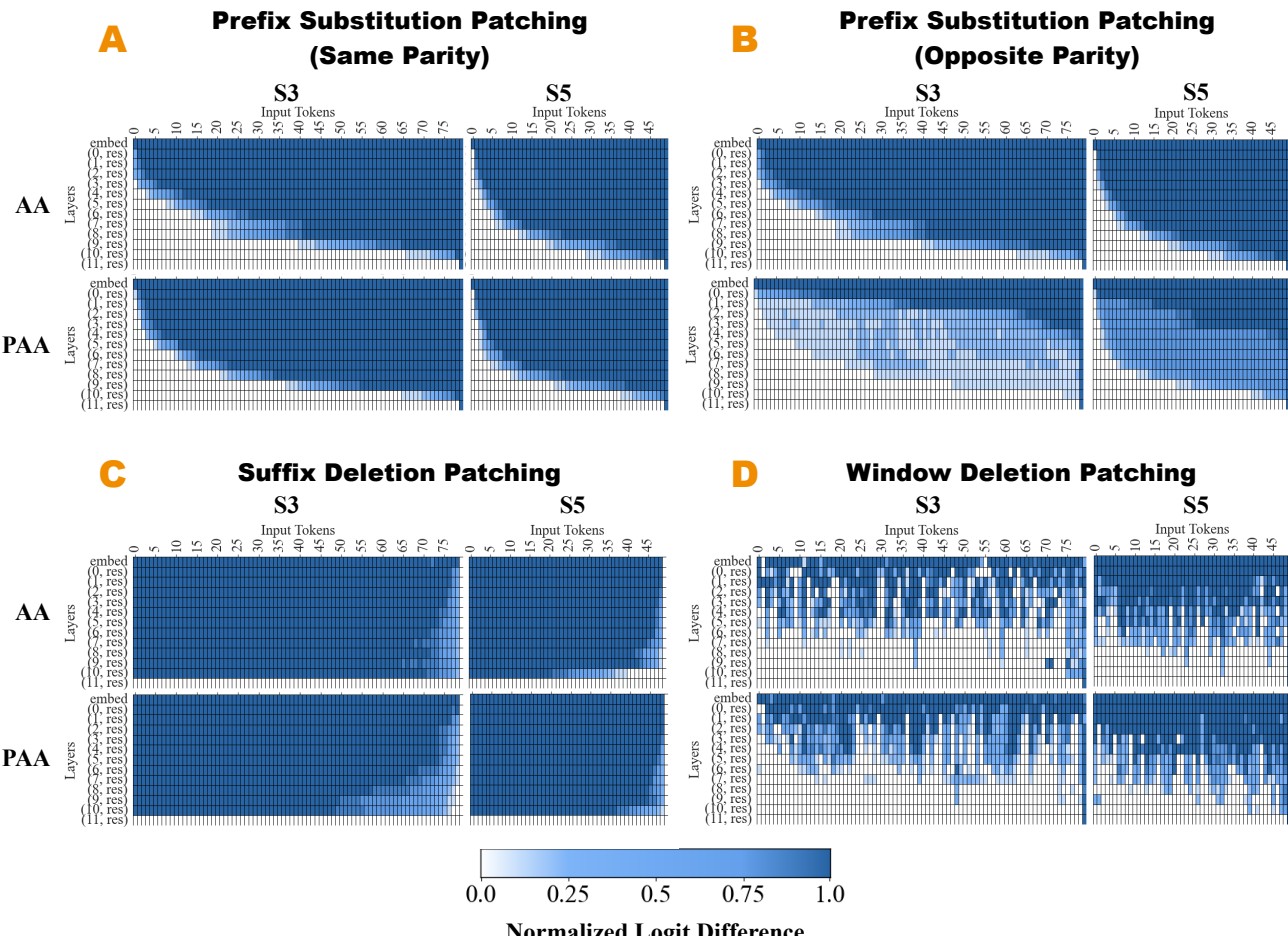

*Figure 8.* Activation patching results across different types of localization (prefix, suffix, window patching) and different types of patching content (substitution, deletion). **(A)**: Prefix-substitution results on only sequences with the *same* parity. We see the same exponential patching pattern in both AA and PAA models, showing that parity complements are computed in the same associative manner. **(B)**: Prefix-substitution results on only sequences with *opposite* parities. We see the exponential patching pattern in AA models, meaning that in AA models, parities are computed with the state. In PAA models, however, the patching pattern is roughly parallel, meaning parities are computed roughly in parallel in early layers. **(C)**: Suffix-deletion results show that we can ignore progressively longer sequences of suffixes as we go down the layers of the network, consistent with how we believe AA and PAA work. **(D)**: Window-deletion results show that important activations are arranged hierarchically, again consistent with how we believe AA and PAA work.

**Increasingly longer suffixes are ignored for AA and PAA models in later layers** In Figure 8C, we show suffix deletion patching results, finding that we can swap out *exponentially longer* both AA and PAA models without affecting the prediction. This is in line with how the associative algorithm in either model works: suffixes of progressively longer lengths are collected into the final token as we go down the layers.

**Important activations are arranged hierarchically** In Figure 8D, we show window deletion patching results, with a window size of 1. We find a patching pattern consistent with the associative algorithm: deleting any single token in the early layers is extremely important, but the spacing of important tokens gets sparser as we go down the layers, consistent with the depiction of AA/PAA in Figure 1. At the bottom layers, deleting any single token is unimportant for the final computation of the state.

**Patching signatures are relatively consistent across examples** In Figure 9, we plot the standard deviations across 200 pairs of S3 inputs for the following three sets of results: (A) prefix substitution patching of AA models, (B) prefix

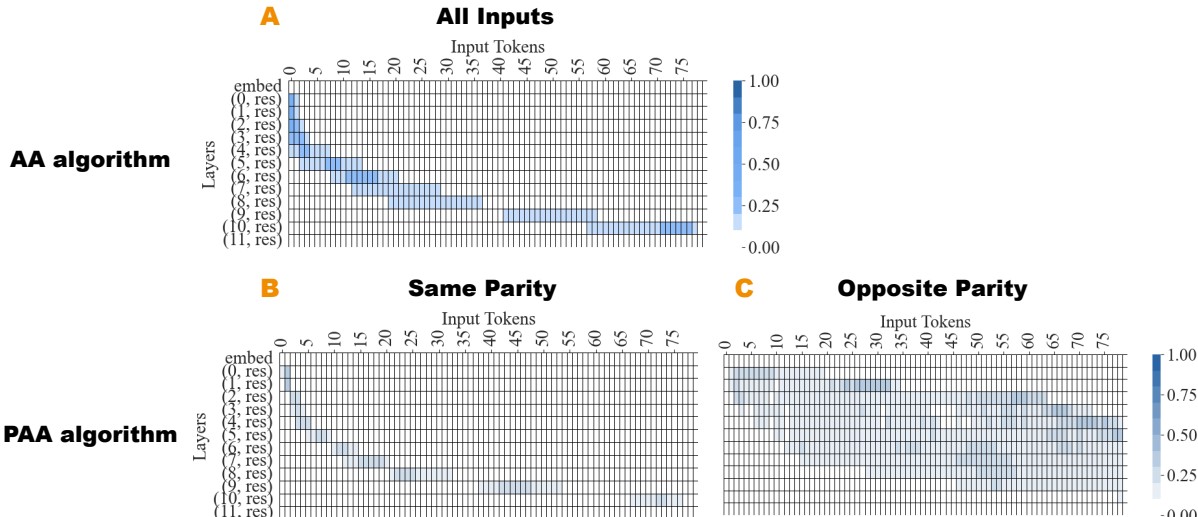

*Figure 9.* Standard deviations of the prefix substitution activation patching results across 200 S3 input pairs, for **(A)** AA models, **(B)** PAA models on pairs with the same parity, and **(C)** PAA models on pairs with opposite parity. We find generally low standard deviations across examples.

substitution patching of PAA models (opposite parity inputs), (C) prefix substitution patching of PAA models (same parity inputs). We find relatively low standard deviations in all three cases, showing that these signatures hold across different examples.

## C. Full Probing Results

### C.1. Probing signatures are relatively consistent across examples

We investigate the sensitivity of our probing signatures to the data on which the probe was trained. We focus on Pythia models trained on S3. For each model type (PAA vs. AA) and each probe type (parity vs. state probe), we train 10 different probes across 10 different random subsets of the S3 dataset, and report the standard deviations of their accuracies across the 10 runs. Results can be found in Table 1. We find low standard deviations (less than $10^{-3}$) in all four cases, indicating that our probe signatures are robust to different subsets of the data and to randomness in probe training.

### C.2. Probe accuracies over sequence lengths

How do the probe accuracies in Figure 3 decompose over sequence lengths? We sweep over $S_3$ sequences $a_1 \ldots a_i$ of lengths ranging from $i = 5$ to 100, and train a linear probe that takes in input $h_{t,l}|a_1 \ldots a_t$ —the layer-$l$, position-$t$ representation of the model on input sequence $a_1 \ldots a_i$ with $t < i$ — and aims to predict the final state $s_i$ from the hidden representation.

The mean probability the probe put on the correct answer is plotted in Figure 10. We find that, generally speaking, both AA and PAA linearly encode states of exponentially longer sequences as they go down the layers. We find evidence that the PAA models use their intermediate layers to compute parity in parallel: at around the second residual layer, PAA models place $\frac{1}{3}$ probability on the correct answer (there are three actions of each parity in $S_3$).

### C.3. Examples of Associative Algorithm Representations that Do or Do Not Linearly Encode Parity

As shown in Figure 1, models that learn AA sometimes encode parity linearly at the final layer but sometimes do not. The examples shown in Figure 3 all do *not* linearly encode parity at the final layer. We show a 3D visualization of the $S_3$ AA model's final hidden representations along the three principal components of the representation (which explain $41.8\%$ of the variance in the data) in Figure 11. As we can see, parity is *not* linearly encoded at the final layer. In Figure 12, we show

| Layer | Pythia on S3 (PAA) | | Pythia on S3 (AA) | |
| | State Probe | Parity Probe | State Probe | Parity Probe |
|---|---|---|---|---|
| 0 | $4.16 \times 10^{-6}$ | $3.03 \times 10^{-6}$ | $6.33 \times 10^{-6}$ | $2.71 \times 10^{-6}$ |
| 1 | $3.36 \times 10^{-6}$ | $3.31 \times 10^{-6}$ | $2.62 \times 10^{-6}$ | $2.82 \times 10^{-6}$ |
| 2 | $4.52 \times 10^{-6}$ | $5.45 \times 10^{-6}$ | $1.46 \times 10^{-6}$ | $8.97 \times 10^{-7}$ |
| 3 | $4.31 \times 10^{-5}$ | $5.59 \times 10^{-4}$ | $5.00 \times 10^{-6}$ | $1.38 \times 10^{-6}$ |
| 4 | $1.58 \times 10^{-5}$ | $3.72 \times 10^{-5}$ | $4.16 \times 10^{-6}$ | $1.11 \times 10^{-6}$ |
| 5 | $1.70 \times 10^{-5}$ | $5.38 \times 10^{-6}$ | $1.97 \times 10^{-6}$ | $1.11 \times 10^{-6}$ |
| 6 | $1.87 \times 10^{-5}$ | $7.02 \times 10^{-6}$ | $4.73 \times 10^{-6}$ | $1.59 \times 10^{-6}$ |
| 7 | $1.29 \times 10^{-5}$ | $1.34 \times 10^{-5}$ | $7.79 \times 10^{-6}$ | $3.05 \times 10^{-6}$ |
| 8 | $3.18 \times 10^{-5}$ | $3.08 \times 10^{-5}$ | $1.18 \times 10^{-5}$ | $2.64 \times 10^{-6}$ |
| 9 | $2.96 \times 10^{-4}$ | $6.85 \times 10^{-5}$ | $2.36 \times 10^{-5}$ | $3.53 \times 10^{-6}$ |
| 10 | $3.10 \times 10^{-4}$ | $7.98 \times 10^{-5}$ | $2.60 \times 10^{-5}$ | $2.96 \times 10^{-6}$ |
| 11 | $1.86 \times 10^{-4}$ | $8.22 \times 10^{-5}$ | $8.83 \times 10^{-6}$ | $2.77 \times 10^{-6}$ |
| 12 | $7.05 \times 10^{-5}$ | $4.01 \times 10^{-5}$ | $1.20 \times 10^{-6}$ | $2.75 \times 10^{-6}$ |

*Table 1.* Standard deviations across probe accuracies. We focus our analysis on S3 Pythia models and train 10 probes on different subsets of the S3 dataset. Standard deviations are tiny in all cases, indicating robust signatures.

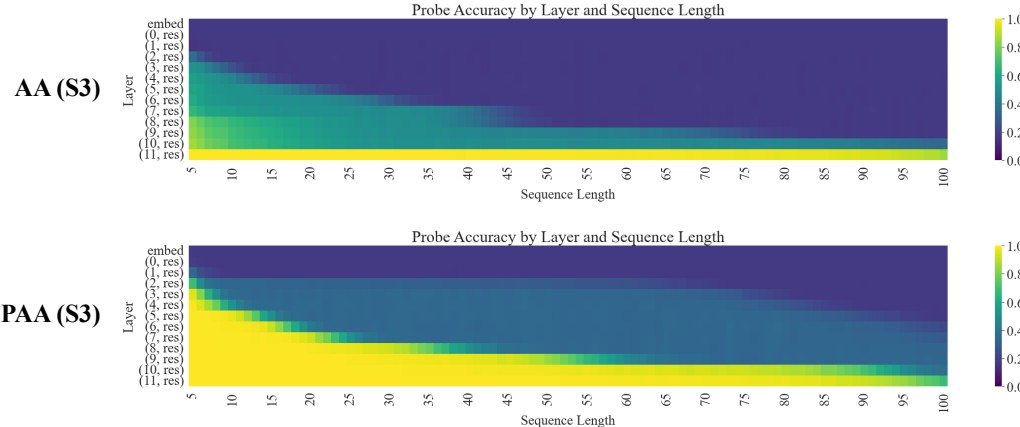

*Figure 10.* We plot the average accuracy of a linear probe trained to predict the final state of an action sequence $A$, given the corresponding final-token hidden representation of AA and PAA models on $A$. We find that both types of models can handle longer sequence lengths as we go down the network, and that PAA models compute the parities of sequences at roughly layer 2, after which they can get the parity of the state correct but not the exact state.

final-layer hidden representations from an AA model that *does* linearly encode the parity (from-scratch GPT2-base on $S_3$). When projected onto three components that explain 49.9% of the variance in the data, we find a clear linear separation between the odd and even parity representations.

## D. Full Linear Decomposition Results

### D.1. $S_3$

We visualize the linear decomposition of the last-layer or penultimate-layer representations across various PAA models. We find the triangular prism shape similar to Figure 4 in all of them, but there was no consistency in which states were paired to form the clusters.

One interpretation is that PAA models may be learning various presentations of $S_3$, with each clustering configuration corresponding to a different presentation. Generally speaking, $S_n$ can be generated by a 2-cycle and an $n$-cycle: any permutation of $S_n$ can be created by composing these two permutations. For example, $S_3$ can be generated by the 2-cycle $1 \leftrightarrow 2$ and 3-cycle $1 \rightarrow 2 \rightarrow 3 \rightarrow 1$, which corresponds to the clustering $\{(123, 213), (312, 132), (231, 321)\}$: the states

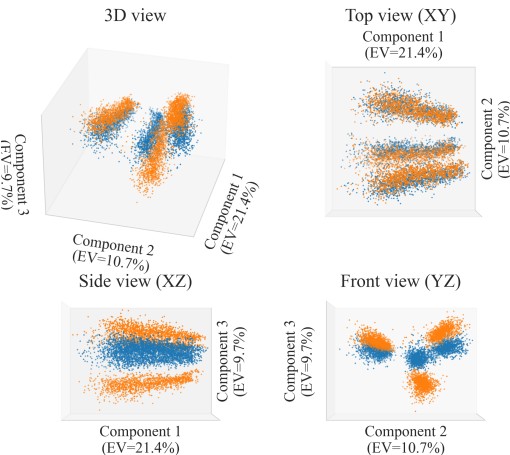

*Figure 11.* Example activations from an AA model that *does not* linearly encodes parity at the final layer, projected on three principal components with a total explained variance of 41.8%. Blue points have even parity, while orange points have odd parity.

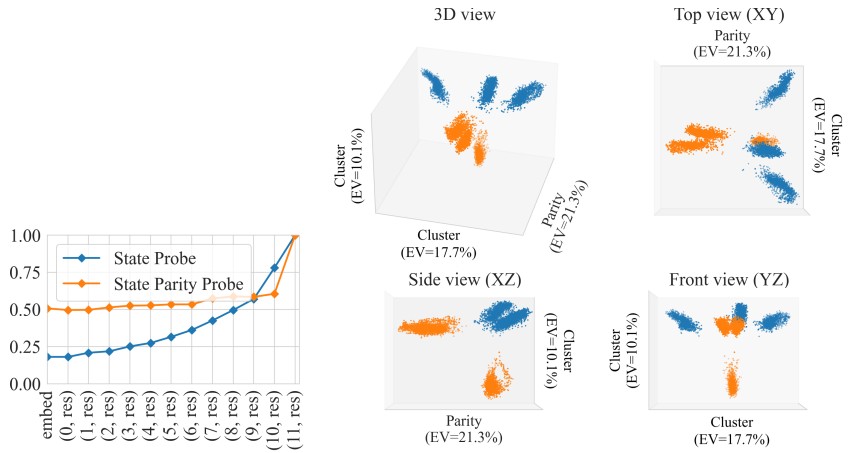

*Figure 12.* Example activations from an AA model that *does* linearly encode parity at the final layer. Blue points have even parity, while orange points have odd parity. (Left) State and state parity probe signatures of this model. (Right) projection of hidden representations onto three components with a total explained variance of 49.9%.

within the cluster can be transformed into each other by applying $1 \leftrightarrow 2$, while states between clusters are related to each other by $1 \rightarrow 2 \rightarrow 3 \rightarrow 1$. PAA models that cluster according to this pattern may have learned these generators.

### D.2. $S_5$

What happens in models that learn PAA in $S_5$? We visualize the penultimate-layer representation of a Pythia-160M model that learned PAA on $S_5$ in 3D space, with parity along one axis and two orthogonal directions along the other two. We find 4 distinct clusters, corresponding to the position of 1 in the state (states having 1 in position 4 and 1 in position 5 are clustered together).

## E. Simulating State Tracking in Natural Language

To emulate a more practical scenario, we train pre-trained and from-scratch Pythia models on a version of the S3 permutation composition task expressed in *natural language*. For example, the permutation "132" would be expressed as "swap positions 2 and 3," while "312" would be "rotate the last item to the front." We train LLMs to predict the final state (e.g., 231) from the final *period* token of the sequence. For example, the following sequence:

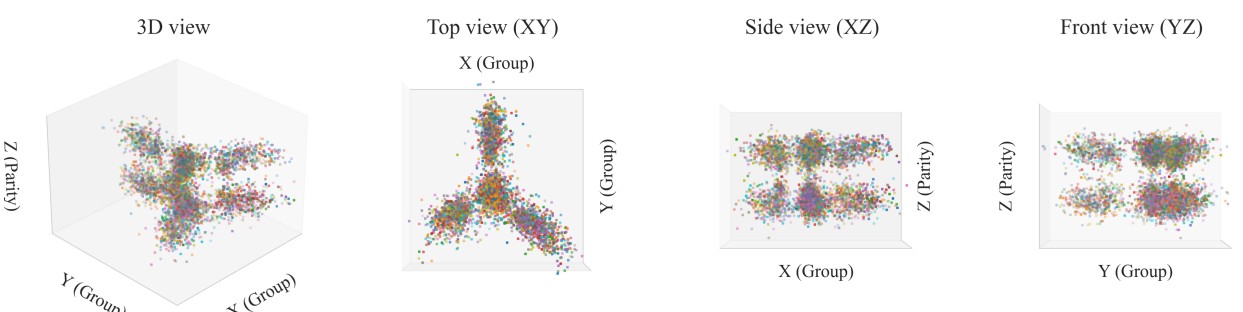

*Figure 13.* Projecting models that learn PAA on $S_5$ into 3D space. Unlike PAA models on $S_3$ (Figure 4), the state cannot be fully represented by a clean decomposition into 3 directions (notice the colors superimposed on each other). However, we do still find symmetry across the parity axis, similar to $S_3$. Moreover, there are 4 neat clusters, one at the center, and three outward protruding "prongs". We find that the clusters correspond to the position of 1 in the state.

> Swap positions 2 and 3. Rotate the last item to the front. Swap positions 1 and 2.

would map to permutation sequence "132, 312, 213" and finally correspond to the state "123" after the swaps. We then conduct a similar style of probing and activation patching experiments on models trained on this task.

### E.1. Probing experiments

We train probes to map from the activation of each layer at the position of the final "." token to the final state. Shown in Figure 14, as in our results on synthetic data, probing results are consistent with associative mechanisms – the state probe improves exponentially over layers. In pre-trained Pythia, we see the model represent the state parity much earlier (in terms of layer) than the actual state representation, a signature of the PAA algorithm. For non-pre-trained Pythia, the accuracy of the state and state parity probes increases at a similar rate, indicating that it is more likely learning an associative algorithm; whether it is the strict AA algorithm we identified in the synthetic case is unclear.

### E.2. Activation patching experiments

We patch prefixes up to a fixed token position.[8] As shown in Figure 14, both models display a distinctly associative signature with exponentially longer prefixes being disregarded for the final state prediction as depth increases. Furthermore, the pre-trained Pythia model possesses a light blue middle section – a sign of the PAA algorithm. Interestingly, the pre-trained Pythia results are significantly more "compressed" over the layers – the LLM computes the state very early on. We suspect this may be due to the pre-trained LLM taking advantage of its innate natural language understanding (and perhaps pre-trained state tracking abilities!) to quickly solve the task in an early layer.

## F. Full Attention Heads Analysis

### F.1. Formalizing Parity Heads

To formalize a metric for whether an attention head behaves like a parity head, we define a *parity head score* as the percentage of sequence lengths (ranging from 5 to 80) over which the head places *significantly* more attention on odd-parity permutations than even-parity permutations, measuring significance using a 95% confidence interval.

**Definition F.1.** Let $\alpha_{i,\ell}^{(H)}(x)$ be the attention weight of the $H$th attention head in layer $\ell$ at position $i$ for input $x$ where $x$ is the list of actions $[a_1 \ldots a_t]$.

---

[8]Note that token positions may no longer be aligned in natural language: "swap" actions have 5 tokens, and "rotate" actions have 7. Thus, we may not be replacing the activations of the same number of actions between prompts. Nonetheless, we still believe the activation patching results serve as a good proxy for estimating how information gets propagated through the layers.

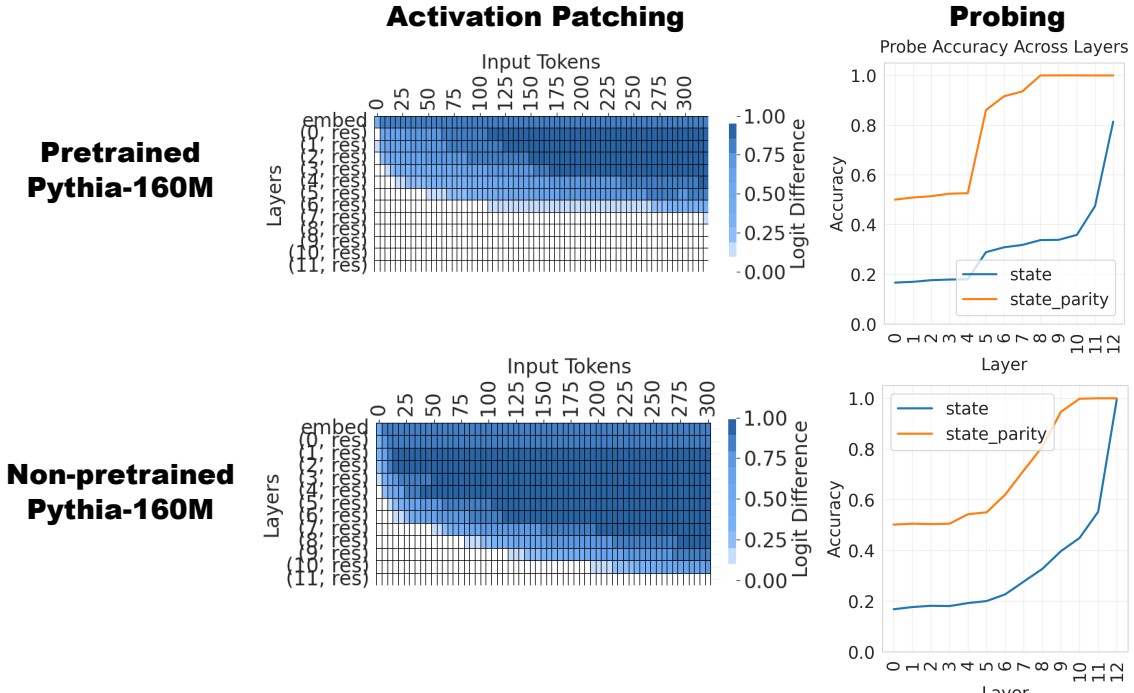

*Figure 14.* Patching and probing result for Pythia models trained on *natural language* permutation composition task. We plot the signature of a pre-trained Pythia model (top) and a non-pre-trained Pythia model (bottom). In both cases, the signatures are consistent with the state being learned associatively (both the patching signature and state probe have an exponential curve). The signature of the pre-trained model is consistent with a PAA signature, with the parity probe converging in early layers, and the activation patching signature containing a light-blue middle section.

Define the sets of attention weights on odd and even tokens of $x$ as:

$$\mathcal{A}_{\ell,H,\text{odd}}(x) = \{\alpha_{i,\ell}^{H}(x) : a_i \text{ is odd}\},$$

$$\mathcal{A}_{\ell,H,\text{even}}(x) = \{\alpha_{i,\ell}^{H}(x) : a_i \text{ is even}\}.$$

We find heads that respond to parity can be local: for example, attention head 5 in layer 1 responds to odd-parity permutations in the midpoint of the sequence, while attention head 4 responds to ones late in the sequence.

Thus, we record the **parity head score** $\sigma_{\ell,H,\text{parity}}$, which measures the *proportion* of the sequence for which more attention is placed on odd-parity actions compared to even-parity actions:

$$\sigma_{\ell,H,\text{parity}} = \frac{1}{L-5} \cdot \sum_{t=5}^{L} \sigma_{\ell,H,\text{parity},t}, \qquad \text{where}$$

$$\sigma_{\ell,H,\text{parity},t} = \mathbf{1}\left(\mathbb{E}[\mathcal{A}_{\ell,H,\text{odd}}([a_1 \ldots a_i])] - 0.95 \cdot \text{CI}_{\ell,H,\text{odd}}([a_1 \ldots a_i]) > \mathbb{E}[\mathcal{A}_{\ell,H,\text{even}}([a_1 \ldots a_i])]\right).$$

Here, $[a_1 \ldots a_i]$ denotes the first $i$ elements of the sequence $x$, and $\text{CI}_{\ell,H,\text{odd}}$ refers to the confidence interval around the average attention weights on odd-parity tokens. We use sequence lengths of up to $L = 80$ for $S_3$ and $L = 50$ for $S_5$.

We find no evidence of parity heads in any layer of AA models. However, we find at least two attention heads with $\sigma$ significantly exceeding 50% in the first few layers of PAA models, highlighted in Table 2.

### F.2. AA Attention Patterns

What sorts of attention patterns appear in AA models? Because attention is dense, we visualize only the top-K attention traces from and to each position at each layer. Specifically, we plot the attentions of an LM on an input as a graph with:

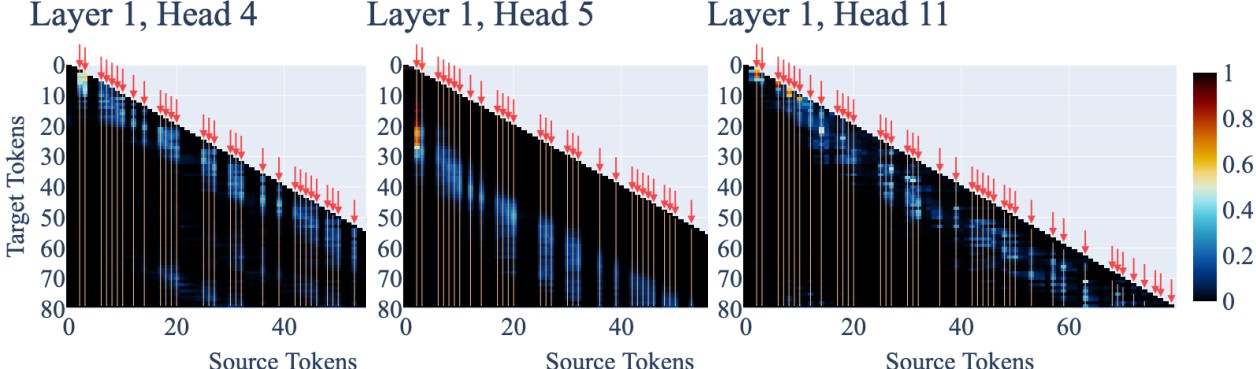

*Figure 15.* Examples of parity heads in a PAA model. We plot heatmaps showing attention weights on each source token at each target token location (we show up to only 60 source tokens for heads 4 and 5 for the sake of space). We draw arrows / yellow lines at source tokens corresponding to *odd-parity actions*. Note that parity heads attend almost exclusively to odd-parity tokens.

| Algorithm | Layer | Head | Parity Head Score |
|---|---|---|---|
| $S_3$ (PAA) | 1 | 1 | $90.1\%_{\pm 3.0\%}$ |
| | 1 | 4 | $86.4\%_{\pm 8.2\%}$ |
| | 0 | 5 | $67.1\%_{\pm 5.1\%}$ |
| $S_3$ (AA) | 0 | 4 | $3.3\%_{\pm 4.5\%}$ |
| | 2 | 7 | $3.0\%_{\pm 6.1\%}$ |
| | 11 | 0 | $2.0\%_{\pm 4.0\%}$ |
| $S_5$ (PAA) | 3 | 3 | $83.6\%_{\pm 9.6\%}$ |
| | 3 | 2 | $80.6\%_{\pm 6.2\%}$ |
| | 2 | 6 | $50.3\%_{\pm 22.4\%}$ |
| $S_5$ (AA) | 0 | 7 | $5.9\%_{\pm 8.2\%}$ |
| | 0 | 3 | $3.8\%_{\pm 5.4\%}$ |
| | 0 | 0 | $3.2\%_{\pm 4.3\%}$ |

*Table 2.* Top-3 parity head scores across all attention heads in each type of model. We report average parity head scores (%) over 100 examples, as well as their standard deviations. Informally, this metric captures the proportion of the sequence over which more attention is placed on odd-parity actions than even-parity actions.

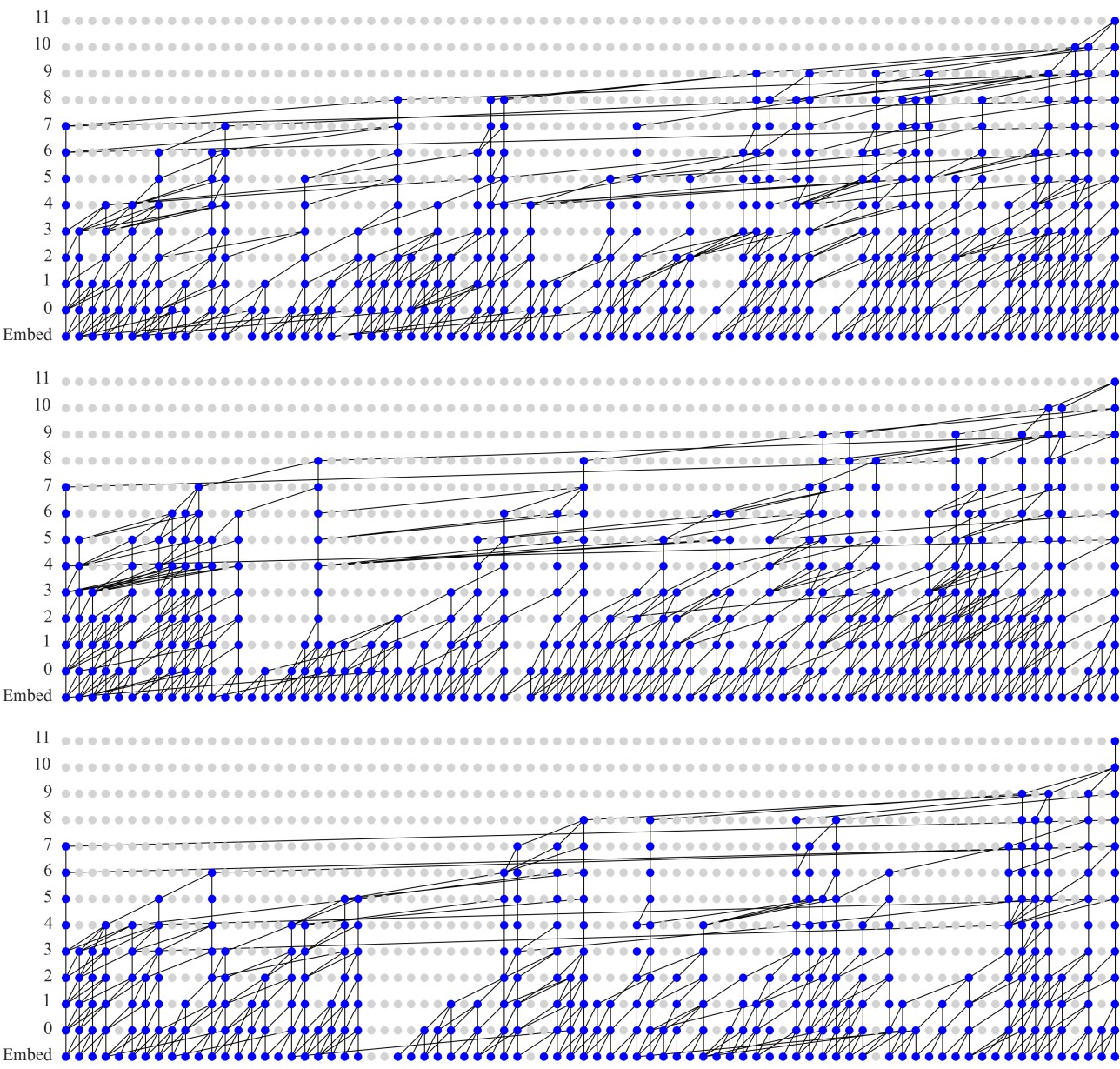

*Figure 16.* Attention patterns in AA models form a tree-like pattern, with tokens in successive layers attending to larger windows of downstream tokens. This is in line with how we expect the associative algorithm to function. We plot only the most salient attention weights by pruning edges for which the attention weight is < 0.95, the target token is not in the top-3 attended-to tokens from the source token, or the source token is not in the top-10 attended-from tokens for the target token. We expect that the attention patterns visualized here do not form a single clean tree, but the superimposition of multiple trees.

1. Nodes $(t, l)$ for each token position $t$ and layer $l$,

2. Edges between two nodes $(t_1, l-1)$ and $(t_2, l)$ if position $t_2$ at layer $l$ attends to position $t$ at layer $l-1$. We define "attends to" as follows:

   Let $\alpha_{t_1 \to t_2}^{(l)}$ denote the maximum attention weight (across all attention heads at layer $l$) from position $t_1$ at layer $l-1$ to position $t_2$ at layer $l$.

   We say $(t_2, l)$ **attends to** $(t_1, l-1)$ if all of the following conditions are met:

(a) $\alpha_{t_1 \to t_2}^{(l)} > 0.95$

(b) $\alpha_{t_1 \to t_2}^{(l)} \in \text{top-3}(\{\alpha_{i \to t_2}^{(l)} : i \in [1, n]\})$: $t_1$ is among the top-3 attended-to tokens for $t_2$ at layer $l$

(c) $\alpha_{t_1 \to t_2}^{(l)} \in \text{top-10}(\{\alpha_{t_1 \to j}^{(l)} : k \in [1, n]\})$: $t_2$ is among the top-10 attended-from tokens for $t_1$ at layer $l - 1$

We show example attention patterns for an AA model on three sample prompts in Figure 16. We only plot the attention subgraph directly connected to the final token position at the final layer (which is used to predict the state). We find that attention in AA models forms a tree-like pattern where successive layers attend to wider and wider context windows, with nodes that are more and more spaced apart. This is in line with how we believe the associative algorithm works: adjacent pairs of actions are grouped together at each layer in a hierarchical manner.

Note that the tree is not entirely clean: there are redundant edges and edges that cross over each other. We suspect that the subgraph we've picked up on is not a single tree, but rather the superimposition of *multiple* trees, each potentially contributing to not just the prediction for the final token, but also the predictions of the previous tokens.

## G. How are these algorithms learned over the course of training?

We conduct a more detailed analysis of the training phases for two Pythia models trained on the $S_5$ task: one that learned AA and one that learned PAA. The training curves are shown in Figure 6, and we investigate the generalization behavior at different points along these curves. Both models improve over training by progressively generalizing to longer sequence lengths, rather than making uniform gains across all lengths. In the case of the PAA model, convergence appears to occur in two distinct phases: first, the model learns the parity of states across the entire length-100 sequence, followed by learning how to predict the state. By contrast, the AA model learns to generalize parity and state simultaneously.

## H. Additional Factors Influencing Learned Algorithm

### H.1. Model Size

We have investigated whether model size influences which algorithm the models learn to implement. As shown in Figure 17, model size empirically does not seem to have much effect on the choice of learned algorithm, with model architecture and initialization having a much bigger effect. Some notable differences between the GPT-2 and Pythia architecture are the use of rotary embeddings, parallelized attention, and feedforward layers rather than sequential, and untied embedding and unembedding.

### H.2. Topic Modeling

The topic model used in Section 5.3 is parameterized as follows: We generate the distribution of the 4 topics in each document using a random Dirichlet distribution with $\alpha = 0.3$. The distribution $p(\text{token} \mid \text{topic})$ for each token $123, 132, 213, 231, 312, 321$ is:

$$\begin{bmatrix} 3.06 \cdot 10^{-2}, & 1.11 \cdot 10^{-1}, & 5.79 \cdot 10^{-4}, & 6.45 \cdot 10^{-3}, & 6.58 \cdot 10^{-3}, & 8.45 \cdot 10^{-1} \\ 3.36 \cdot 10^{-1}, & 1.69 \cdot 10^{-4}, & 6.63 \cdot 10^{-1}, & 8.05 \cdot 10^{-7}, & 1.68 \cdot 10^{-7}, & 7.81 \cdot 10^{-4} \\ 7.92 \cdot 10^{-5}, & 1.41 \cdot 10^{-2}, & 9.44 \cdot 10^{-1}, & 4.53 \cdot 10^{-4}, & 4.13 \cdot 10^{-2}, & 3.27 \cdot 10^{-11} \\ 2.85 \cdot 10^{-3}, & 1.29 \cdot 10^{-9}, & 7.06 \cdot 10^{-1}, & 6.37 \cdot 10^{-7}, & 2.58 \cdot 10^{-3}, & 2.89 \cdot 10^{-1} \end{bmatrix}$$

We also trained LMs using a topic model with a second token-topic distribution, aiming to distinguish the effect of this particular topic distribution from the effect of topic modeling pretraining in general. On the second distribution, we also find that both randomly initialized GPT-2 and Pythia models learn AA in Figure 18. The $p(\text{token} \mid \text{topic})$ distribution for this model is listed below:

$$\begin{bmatrix} 9.31 \cdot 10^{-1} & 2.32 \cdot 10^{-3} & 1.38 \cdot 10^{-8} & 5.86 \cdot 10^{-10} & 4.62 \cdot 10^{-5} & 6.63 \cdot 10^{-2} \\ 2.10 \cdot 10^{-4} & 3.12 \cdot 10^{-4} & 9.32 \cdot 10^{-7} & 9.07 \cdot 10^{-6} & 2.53 \cdot 10^{-1} & 7.47 \cdot 10^{-1} \\ 4.95 \cdot 10^{-1} & 1.18 \cdot 10^{-3} & 4.55 \cdot 10^{-1} & 2.17 \cdot 10^{-2} & 1.86 \cdot 10^{-8} & 2.71 \cdot 10^{-2} \\ 6.55 \cdot 10^{-1} & 4.92 \cdot 10^{-4} & 3.44 \cdot 10^{-1} & 2.28 \cdot 10^{-7} & 1.94 \cdot 10^{-4} & 2.14 \cdot 10^{-8} \end{bmatrix}$$

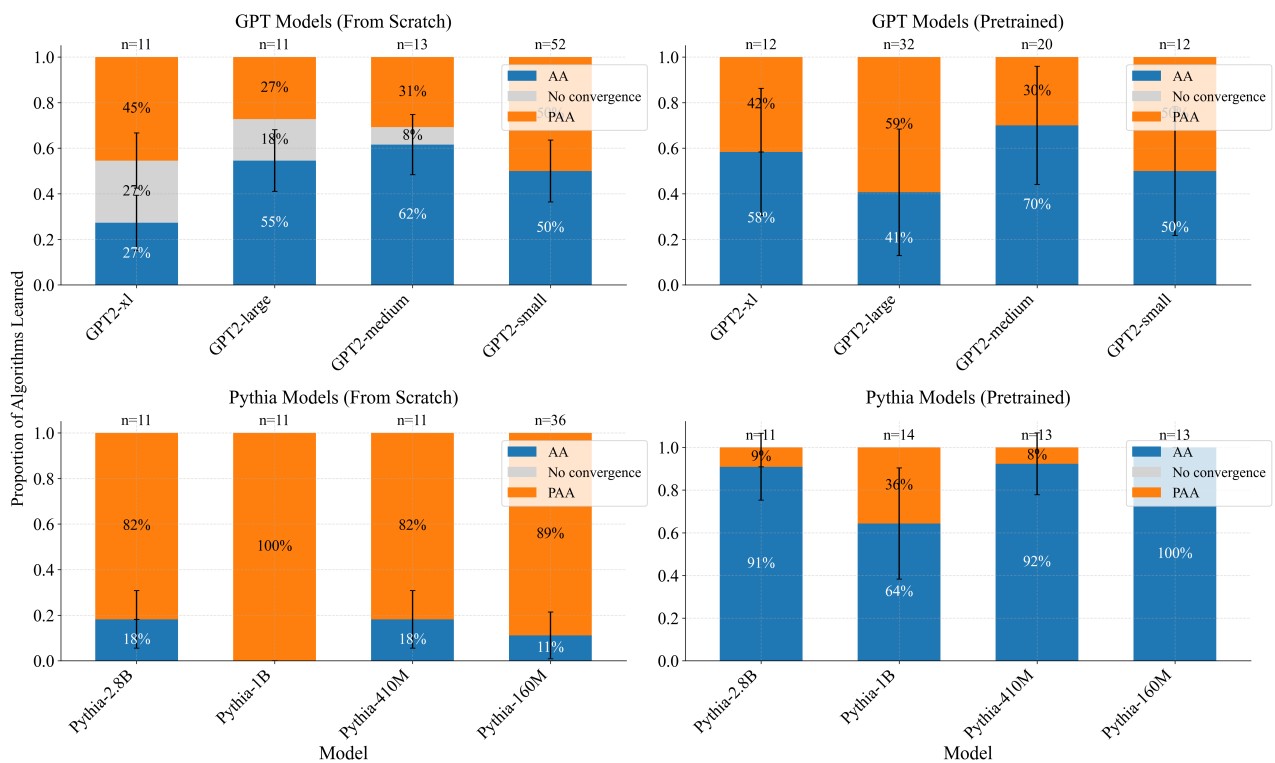

Figure 17. Proportion of $S_3$ algorithms learned by models of various sizes from the GPT-2 and Pythia families. Model architecture and initialization are much bigger factors in influencing the algorithms learned than model size.

## H.3. Parity Loss Curriculum

We explored an additional procedure that encourages models to learn parity via an extra loss term. We train an extra linear classifier that takes in the residual activations of an early layer (e.g., layer 3) in which we find parity to be typically computed through probing. We train with this additional loss term to induce the representation to linearly encode parity early on. The classifier is trained to output (1) parity or (2) a (parity, action) tuple, to ensure that the residual also encodes the original action, and not just the parity. After training, we evaluate the model on the original $S_3$ task. Both procedures induce the model to learn PAA consistently, though the (parity, action) classifier typically allows the model to generalize better.

## H.4. Length Curriculum

We implemented a curriculum training approach where the model was progressively exposed to documents of increasing length. We first trained on only the initial 10 tokens, then expanded to 25 tokens, 50 tokens, and finally the complete 100-token sequences. Each stage of the curriculum was trained for a fixed number of epochs (data). The goal of such a curriculum is to push the model to learn the associative algorithm (AA), as the parity heuristic, we hypothesize, might be less useful for shorter sequences. However, empirically, such a curriculum has no obvious effect on the kind of algorithm the model learns. Of the 5 trials, 2 trials of GPT-2 learn AA, and the other 3 trials learn PAA.

We discovered that all five models can perfectly generalize both parity and state when trained on the first 25 tokens. It is only when trained to generalize to sequence length 50 that the distinction between PAA and AA emerges. Models learning the PAA algorithm do not generalize, while models learning AA can generalize from length 25 to 50, as illustrated by Figure 19. This further confirms our finding that the model learns the $S_3$ algorithm early on.

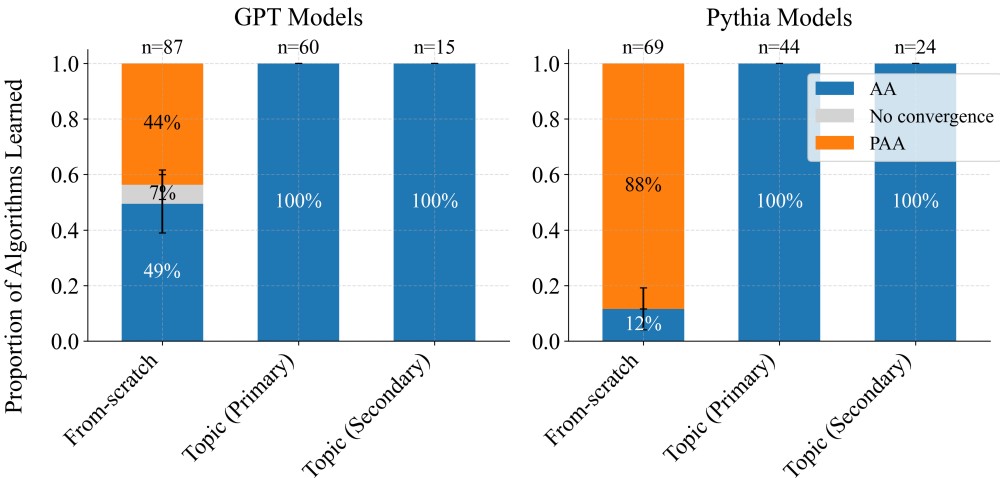

*Figure 18.* Proportion of $S_3$ algorithms learned by models first pre-trained on the two topic models compared to the randomly initialized baselines. Topic modeling, regardless of the specific distributions, pushes the model to learn AA.

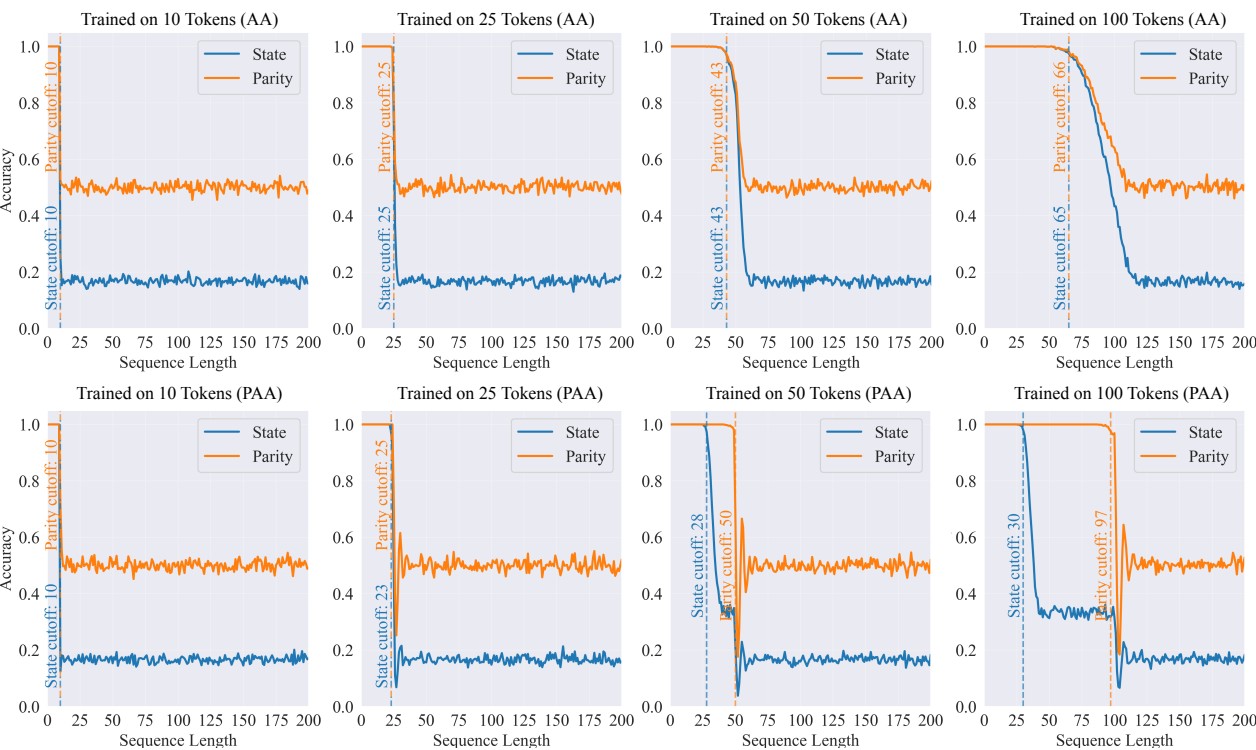

*Figure 19.* Model generalization curves when trained using a length curriculum after training on 10, 25, 50, and 100 tokens, respectively. While the length curriculum doesn't push the model to learn one algorithm or the other, it shows that the model learns these algorithms early on, as indicated by whether the model can generalize well from training on 25 tokens to 50 tokens.

