# OpenReview forum: "(How) Do Language Models Track State?"
_ICML.cc/2025/Conference — ICML 2025 poster_

### Official Review · Reviewer_nf2n · 2025-02-23

**Overall Recommendation:** 3

**Summary:**

This paper investigates state tracking mechanisms in transformer based language models. In particular, this paper applies interpretability analyses to Pythia and GPT-2 models as they solve the word problems in $S_3$ and $S_5$. The authors find signatures relating to two computational models, PAA and AA, and demonstrate that different training curricula can give rise to models using these two mechanisms.

**Claims And Evidence:**

The authors provide strong evidence that the signatures they propose are realized in the transformers they analyze.

**Essential References Not Discussed:**

I was a bit surprised to read the second sentence of this paper
> A growing body of work suggests that these models learn to represent the latent state of the world

These papers would seem to argue differently
* Merrill and Sabharwal '23, "The Parallelism Tradeoff: Limitations of Log-Precision Transformers" and Merill et al '22 "Saturated transformers are constant-depth threshold circuits"
* Huet et al '25, "Episodic Memories Generation and Evaluation Benchmark for LLMs"
* Bhattamishra et al '20, "On the Ability and Limitations of Transformers to Recognize Formal Languages."
* Deletang et al 2023
* Sarrof et al 2024
* Strobl et al 2024


A more nuanced discussion was provided in Section 2.1, however.

It would be nice to at least have some references accompanying the second sentence of the paper. And, the authors might consider a more balanced second sentence also acknowledging work that suggests that transformer based LMs can really struggle with tracking state.

**Experimental Designs Or Analyses:**

I did not check the experiments.

**Methods And Evaluation Criteria:**

The proposed methods make sense.

However, I was a bit confused about the pretraining vs training distinction.

When the models are said to be pretrained (cf line 282), does this means sometimes they were LLMs trained on the entire internet, and then other times (Section 5.3) the weights were initialized randomly, and they were pre-trained explicitly on state tracking problems?

**Other Comments Or Suggestions:**

The extra dotted lines in Figure 1 (for the state parity probe) are a bit confusing until one reads the footnote non the bottom of page 5, but this takes quite a while to get to.

**Other Strengths And Weaknesses:**

The paper as a whole is very well written.

**Questions For Authors:**

My essential question is about the correctness of the AA and PAA signatures, see my question under "Theoretical Claims"

**Relation To Broader Scientific Literature:**

There is a lot of discussion about whether or not foundation models can track state, and this paper makes a nice contribution by showing concretely mechanisms by which transformers could track state and empirically verifying them on pretrained models.

**Theoretical Claims:**

I did not follow the rationale for the AA and PAA signatures. This is an important issue, because the empirical results hinge on the AA and PAA signatures actually corresponding to the AA and PAA algorithms. I think it's very important that the authors clarify this important point.

Here are my questions and confusions in detail about the theoretical aspects of this paper.

1. In the algorithm blocks, $n$ is used? What is $n$? Is it the number of tokens? In the description of the word problem (Section 2.2), $t$ is used for the number of tokens. I find this shift in notation very confusing.

2. I do not understand the signatures for AA and PAA as described in this paper. For example, take Figure 1. In the prefix patching signature section for AA, there are four rows of squares, where the first row is entirely blue, and the bottom row only as a blue square in the lower right hand side.

This signature simply does not make sense to me with the definition of prefix patching as described in this paper.

I am understanding the top row to be the input layer, and the bottom row to be a deepest layer, as would seem to be suggested by the diagrams of the algorithms in Figure 1. My understanding is that the token index is increasing from left to right.

So, I don't understand how merely patching the very first token in the first first layer (which seems to be the upper left) could result in an algorithm with a corrupted input predicting the correct state. I would have thought that this clean input would have been corrupted by running the AA algorithm with states corresponding to an entirely different input sequence.

I've thought as hard as I can about this issue and it just doesn't make any sense to me. Unfortunately I can't recommend accepting this paper while this confusion persists, because everything else about this paper seems like a really solid piece of work. I hope that the authors can provide a crystal clear explanation of the PAA and AA signatures during the rebuttal period.

---

> ### Author Rebuttal · Authors · 2025-04-01
>
> We are glad the reviewer found this paper to be a “really solid piece of work” and “as a whole [...] very well written”! We are happy to clarify your questions below:
>
> ## Pretraining vs training distinction, were pre-trained models pre-trained explicitly on state-tracking problems or the entire internet?
>
> When we talk about pre-trained models, we always mean models pre-trained on the entire internet (*not* state-tracking problems), unless otherwise specified. We will clarify this in future versions!
>
> ## What is $n$? Is it the number of tokens?
>
> Good catch! Yes, we meant $t$ instead of $n$ for the number of tokens in the algorithmic blocks. Apologies for the confusion! We will fix this in future versions of our paper.
>
> ## Figure 1: In the prefix patching signature for AA, there are four rows of squares, where the first row is entirely blue, and the bottom row only has a blue square in the lower right-hand side.
>
> This is merely a high-level extrapolation of the empirical result shown in Figure 2, which might make more sense with more details.
>
> ## I am understanding the top row to be the input layer, and the bottom row to be a deepest layer, the token index is increasing from left to right.
>
> This is correct!
>
> ## Why does just patching the first token result in an incorrect state?
>
> Great question! **We create pairs of prompts by corrupting only the *first* token of the prompt**. This is shown visually in section B of Fig 1 (the two prompts differ on action $a_1$), but we realize how this may be easy to miss and we will make it clearer in the final submission. Corrupting only the first token would result in two prompts with a differing final state. Restoring the activation of the first token at the embedding layer is equivalent to prompting the LM with the correct prompt, and thus is sufficient for producing the final correct state.
>
> Having the two prompts differ only at the first token allows us to track the impact of that first token through the network. In early experiments where we corrupt a token near the middle or end of the prompt, we found similar patching patterns, only more vertically compressed (as would be expected).
>
> We will add this clarification to Section 2.3 and Section 4.2 in the paper!
>
>
> ## Is claim “A growing body of work suggests that these models learn to represent the latent state of the world” true in light of other work?
>
> This is a fair point. We want to emphasize the difference between (1) "state representations are decodable" and (2) "models use state tracking procedures that generalize to arbitrarily long inputs." The first has been shown by prior work [2,3], while the second remains underexplored [4] but also is not a necessary condition for (1). We will add the aforementioned references for (1) to the second sentence, and soften the claim of the sentence to the following:
>
> A growing body of work suggests that the latent state of the world can be decoded from model internals—e.g. situations described by language and results of program execution—to support prediction.
>
> We will also add the suggested sources – thanks for providing them!
>
>
> ## The extra dotted lines in Figure 1 (for the state parity probe) are a bit confusing until one reads the footnote on the bottom of page 5
>
> Thanks for noting this! We will add a line in the caption in Figure 1: “Note the dotted lines indicate two different probing signatures that would both be consistent with this algorithm (see Appendix C.1 for more details).”
>
>
> [2] Li et al., 2022. Emergent World Representations: Exploring a Sequence Model Trained on a Synthetic Task. https://arxiv.org/pdf/2210.13382
>
> [3] Li et al., 2021. Implicit Representations of Meaning in Neural Language Models. https://arxiv.org/abs/2106.00737

---

> > ### Comment · Reviewer_nf2n · 2025-04-04
> >
> > Thank you for your response!
> >
> > > We create pairs of prompts by corrupting only the *first* token of the prompt.
> > Thank you for this very helpful clarification. Yes, I think I now understand. I agree it would be extremely helpful to explicitly include this sentence in the main text (as opposed to having to discern this experimental design choice from a figure caption and an inline equation.
> >
> > > A growing body of work suggests that the latent state of the world can be decoded from model internals—e.g. situations described by language and results of program execution—to support prediction.
> >
> > I strongly recommend that you include at least three references cited after the phrase "a growing body of work." Otherwise it's a bit vague and handwavy.
> >
> > # Some additional thoughts and recommendations
> >
> > ## Abstract is vague
> >
> > This sentence of the abstract is vague:
> >
> > > The two mechanisms exhibit markedly different robustness properties, and we show how to steer LMs toward one or the other with intermediate training tasks that encourage or suppress the heuristics.
> >
> > From reading the abstract, I can't tell if AA or PAA is better. In the conclusion you clearly write
> >
> > > LMs that learn the AA algorithm tend to generalize better and converge faster.
> >
> > I would strongly recommend that you replace the vague abstract sentence with this much clearer concluding sentence. Then someone picking up the paper for the first time can see that you are saying AA is more reliable than PAA just from the abstract.
> >
> > ## Figure 1B is hard to read
> >
> > Thank you for the clarification about Figure 1B. Is there any way to make the figure itself clearer, i.e. that the squares are layers and the top is an input layer and the bottom the deepest layer? With words? It's just so hard to tell what's going on.
> >
> > I'm raising my score to a 3. Nice job with this paper!

---

### Official Review · Reviewer_qUEj · 2025-03-12

**Overall Recommendation:** 3

**Summary:**

This paper investigates how language models track dynamic states through systematic analysis of permutation composition problems . The study reveals that both pretrained and fine-tuned Transformer models learn two distinct state-tracking mechanisms: an Associative Algorithm (AA) and a Parity-Augmented Associative Algorithm (PAA). The former solves tasks via hierarchical permutation composition, while the latter first prunes the state space using parity heuristics before refining results with AA. Experimental validation demonstrates the divergence between these mechanisms, and the authors propose intermediate tasks to guide model specialization.

**Claims And Evidence:**

yes

**Essential References Not Discussed:**

N/A

**Ethical Review Flag:**

Flag this paper for an ethics review.

**Experimental Designs Or Analyses:**

yes

**Methods And Evaluation Criteria:**

yes

**Other Comments Or Suggestions:**

N/A

**Other Strengths And Weaknesses:**

### Strengths
1. The permutation composition framework offers theoretical significance and generalizability to practical tasks like finite automata simulation, establishing a concise yet versatile testbed for state tracking research.
2. The work elucidates how architectural choices and initialization parameters influence algorithmic preference, while proposing actionable interventions to optimize state-tracking capabilities.

### Weakness
1. Exclusive reliance on artificially designed permutation tasks (S3/S5 groups) leaves unverified whether observed mechanisms generalize to real-world scenarios like natural language inference or code execution. Despite claims of generalizability, the absence of cross-task validation or concrete application cases substantially weakens practical relevance.
2. While noting that initialization and architecture dictate algorithm preference, the paper fails to uncover root causes(e.g., how initial parameters bias computational pathways or how attention implement parity heuristics). This black-box attribution reduces findings to correlational observations rather than reproducible causal mechanisms.
3. Exclusive focus on final-state accuracy and parity correctness neglects critical dimensions of state tracking, including intermediate state consistency, robustness to long-range dependencies, and error tolerance under adversarial perturbations. Overreliance on parity shortcuts risks overestimating models’ true tracking capabilities.

**Questions For Authors:**

1. The paper mentions that model architecture influences algorithmic preference but lacks concrete analysis of how depth (number of layers) or hidden layer dimensionality (width) modulates the AA/PAA propensity. Do deeper networks inherently favor AA?

2. If applying the mechanisms discovered in permutation tasks to practical scenarios (e.g., tracking dialogue states or program execution traces), would structural modifications or specialized training strategies be required?

**Relation To Broader Scientific Literature:**

N/A

**Theoretical Claims:**

yes

---

> ### Author Rebuttal · Authors · 2025-04-01
>
> Thank you so much for your time and feedback. Below, we address some of your general critiques and specific questions.
>
> ## How could these mechanisms be applied to practical scenarios?
>
> To emulate a more practical scenario, we train Pythia models on a version of our task with permutations expressed in *natural language*, e.g. “132” would be “swap positions 2 and 3.” while “312” would be “rotate the last item to the front.” We train LLMs to predict the *final state* (e.g. 231) from the *final period token* of the sequence. For example,
>
> > Swap positions 2 and 3. Rotate the last item to the front. Swap positions 1 and 2.
>
> would map to state “123”.
>
> We then conduct probing and activation patching experiments.
>
> **Probing experiments**
>
> We train probes to map from the activation of each layer at the position of the final “.” token to the final state.
>
> * Pretrained Pythia-160M (PAA signature): https://ibb.co/5XwfF5h7
> * Non-pretrained Pythia-160M (AA signature): https://ibb.co/R4HfbXhn
>
> As with our results on synthetic data, probing results are consistent with the AA and PAA mechanisms respectively.
>
> **Activation patching experiments**
>
> We patch prefixes up to a fixed token position N. Note that token positions may no longer be aligned in natural language: e.g. “swap” actions have 5 tokens, while “rotate” actions have 7. Thus, we may not be replacing the activations of the *same* number of actions between prompts.
>
> Nonetheless, we still believe the activation patching results serve as a good proxy for estimating how information gets propagated through the layers.
>
> * Pretrained Pythia-160M: https://ibb.co/Tqdcfb4N
> * Non-pretrained Pythia-160M: ​​https://ibb.co/0VXSNNxP
>
> As above, these resemble AA and PAA signatures from the paper. Interestingly, the pretrained results are significantly more “compressed” over the layers – the LLM computes the state very early on. We suspect this may be due to the pre-trained LLM taking advantage of its innate natural language understanding (and perhaps pre-trained state tracking abilities!) to quickly solve the task in an early layer.
>
>
> ## Root causes for why models are biased towards one or the other algorithm
> The goal of this work wasn’t to provide a definitive account of how these algorithms emerged, but rather identify them, provide evidence for how they contrast, and factors that influence how they emerge over training. We believe a comprehensive causal account of the emergence of these algorithms is out of the scope of this paper, and may require novel interpretability tools that have not yet been invented. We hope future research can be conducted on this topic, and we will add a point on this in our conclusion!
>
> ## Exclusive focus on final-state accuracy neglects intermediate state accuracy, robustness to long-range dependencies, and error tolerance under adversarial perturbations
> We respectfully disagree about the first two.
>
> 1. We train and evaluate models to predict all intermediate states up to the sequence length 100 (Section 4.1). We also use probing techniques to verify whether the model's international representation aligns with the ground truth (4.3).
>
> 2. We also examine long-range dependencies by performing activation patching early in the sequence (the very first token!) and seeing how it affects the state prediction at the *last* token (Section 4.2).
>
> We would appreciate clarification on what kind of experiments would best elucidate “error tolerance under adversarial perturbations,” as we're uncertain which adversarial perturbations would be most relevant in the context of our setup.
>
> ## Overreliance on parity shortcuts risks overestimating models’ true tracking capabilities.
>
> We’d appreciate some clarification on how our measurements of the degree to which models rely on parity heuristics affects our estimation of LMs’ state tracking capabilities! For context, our analysis is grounded in whether the model can complete the permutation group task, i.e. their “true tracking capabilities”. We find that models implement an associative scan in both algorithms, and the parity shortcut is simply a heuristic that eliminates incorrect answers in earlier layers for some models (see Sec. 4.2 and Appendix B). In this sense, the parity shortcut cannot implement state tracking on its own and is only complementary to the “true tracking” algorithm. Note that in all our Section 4 analyses (probing, activation patching, generalization, etc.), models are able to achieve 100% accuracy on exponentially longer sequences as we go down the layers, which is impossible if they were relying on a parity heuristic alone.
>
> ## Do deeper/wider networks inherently favor AA?
>
> We do this analysis in Section 5.2 (please check out Figure 6). We found that model size isn’t correlated with the algorithm the model chooses to learn; rather, model architectural and initialization differences matter more.

---

### Official Review · Reviewer_4TFk · 2025-03-12

**Overall Recommendation:** 4

**Summary:**

The paper studies the mechanisms that Transformers learn for performing state tracking - predicting the state after a sequence of operations. Specifically, the model is provided with a sequence of permutations $a_1, \dots, a_t$, and needs to compute the state, which is the composition of the permutations $s_t = a_t \circ \dots \circ a_1$. The authors discuss possible algorithms for performing state tracking, and demonstrate that Transformers learn either an "associative scan" algorithm, which computes in parallel compositions of permutations, or a "parity-associative scan" which uses a computation of the parity of the permutations for ruling out some of the outputs before performing associative scan. The authors discuss the difference between the two mechanisms and how a model can be steered towards performing one mechanism and not the other.

**Claims And Evidence:**

I think the paper is very well-written, and studies a very central question regarding the computational abilities of Transformers and how they learn to solve state tracking problems. The results are novel and interesting, the authors demonstrate that Transformers learn a surprising algorithm for state tracking that leverages the permutation parity computation in order to find the final state. Additionally, the results on steering the model towards one solution instead of the other are very interesting.

Some questions and comments:
1. I think the description of the task is not clear enough:
- Are the permutations provided as individual tokens (i.e., the size of the vocabulary is the number of permutations)?
- "the input to the model is a sequence of actions $[a_1, \dots, a_t]$ and the output is a sequence of state predictions $[s_1, \dots, s_t]$": is this a causal language model trained to perform a sequence-to-sequence task (i.e., not next-token prediction)? Or is this an encoder-only sequence-to-sequence model? What would be the equivalent next-token prediction variant of this task, and do you think this changes the results/conclusions?
2. Related to the previous point: if I understand correctly, the model is provided with the sequence of states after each step as a supervision. This can potentially change the function that the model learns, as it could potentially guide it to generate a particular state at a particular position. Did you try training this end-to-end (i.e., get a sequence of actions and output only the final state)? Would this change the results/conclusions?
3. In Section 3, the description of how the different algorithms are implemented by the Transformer implicitly assumes that everything is implemented starting from the first layer. If the Transformer has more layers than the minimal number of layers required for solving the task, it could potentially pass the tokens through the first layers and start implementing the algorithm deeper in the network, or otherwise skip layers in the middle. Is the claim that the model always learns the most compressed (fewer number of layers) version of the algorithm starting from the first layer?
4. The experiments are done with data repetition (generating 1 million sequences repeated for 20 epochs). Is there a reason for not training on fresh data? Does this change the results?
5. In Figure 5 - what is the sequence length that the models are trained on?
6. In Section 5.3 - do you have any intuitive explanation for why adding the topic modeling steers the model towards the associative scan algorithm?
7. I think that adding an experiment that connects the results in the paper to a more realistic setting (e.g., word problems that require state tracking) would be a nice addition to the paper.

Minor:
- I think that putting labels on the axes of the matrices in Figure 1 (Layers/Sequence Length) will make it easier to parse.
- Typo in line 302 left and 282 right: "S3 and S3" => "S3 and S5".

**Essential References Not Discussed:**

No

**Experimental Designs Or Analyses:**

See above.

**Methods And Evaluation Criteria:**

See above.

**Other Comments Or Suggestions:**

N/A

**Other Strengths And Weaknesses:**

N/A

**Questions For Authors:**

See above.

**Relation To Broader Scientific Literature:**

See above.

**Theoretical Claims:**

See above.

---

> ### Author Rebuttal · Authors · 2025-04-01
>
> We are thankful for your positive feedback and thoughtful comments. Below, we address some specific questions:
>
> ## Are permutations provided as individual tokens  (i.e., the size of the vocabulary is the number of permutations)?
>
> Yes, they are individual tokens! We append them as special tokens to the tokenizer, but since none of the original tokens are used, the vocabulary is effectively the number of permutations. See our response to qUEj (point #1) about results with natural language text rather than these tokens.
>
> ## Clarify training supervision: Is this a causal LM trained to perform a seq2seq task? Is this NTP? Or an encoder-only seq2seq model? What is the equivalent NTP variant?
>
> You can think of this as a *causal decoder-only LM* trained to perform a sequence classification task – for each prefix of a sequence, it is trained to predict the state from the last token of the prefix. Because any action is valid from any state in the permutation composition task, there would be no way to distinguish between the various states through NTP alone. Thus, we did not run an NTP version of the task, though we concur that this would be an interesting avenue of future work.
>
> ## Would training end-to-end to output final state change results/conclusions?
>
> Great question! We chose our original method for a denser supervision signal from each example. We suspect that training end-to-end won’t make a big difference, but would slow training down.
>
> ## Is the claim that the model always learns the most compressed (fewer number of layers) version of the algorithm starting from the first layer?
>
> These Section 3 descriptions are simply meant as high-level guides and illustrations of general algorithms; in practice, LMs may skip layers or tokens, or even use a combination of algorithms. As shown in Figure 14, LMs appear to implement something much more redundant and complex than the illustrated associative algorithm in Figure 1.
>
> ## Why not train on fresh data? Why data repetition? Does this change results?
>
> This is purely for memory reasons. It is hard to check each new data point is unique beyond a certain point, due to needing to store all previously-generated data points in memory. (We ensure all our training examples are distinct from each other, and distinct from the evaluation set.) We use a sufficiently big dataset that we don’t think it’ll make a real difference realistically.
>
> ## Figure 5: seq length the model is trained on?
>
> The model is trained up to sequence length 100, see L280 in the main text. We’ll also include this information in the figure caption in future versions of the paper!
>
> ## Sec 5.3: Intuitively why does topic modelling steer towards associative algorithms?
>
> We don’t have a definitive answer to this question, but here’s a hypothesis: when we're training with topic modeling, the model learns a summary representation of the action tokens that are in the place of the parity scheme the model would normally learn otherwise. Training downstream on state tracking, It would be hard for the model to unlearn the topic modeling representation and pick up on the parity heuristic representation, rather than adapting the existing representation to solve the state tracking task.
>
> ## Adding an experiment with a more realistic setting
>
> Great question! See our response to qUEj (point #1).
>
> ## Minor typos
>
> We will fix, thank you!

---

### Official Review · Reviewer_jeEg · 2025-03-14

**Overall Recommendation:** 4

**Summary:**

It is known that transformers are theoretically able to capture certain formal language tasks of length $n$ with depth $\log(n)$. Empirically, large language models, which are primarily based on transformers, do appear to learn to state track. A full understanding of the mechanism that they learn for state-tracking is still missing. This paper analyses language models trained on permutation word problems (which for a dictionary of greater than 5 is $NC^1$-complete).

Four possible mechanisms are hypothesised, along with what patterns would be expected from prefix patching and probing experiments. This paper shows that the language models tested consistently learn one of two state tracking mechanisms for this task. Generalisation properties of these mechanisms are explored and how to steer the network to converge from one mechanism to the other.

**Claims And Evidence:**

This paper proposes 4 possible schemes that language models can learn to state track and illustrates the expected signatures for the probing and patching experiments. The extensive experimental results match the signatures expected from two of these four schemes.

The signatures are necessary for the AA and PAA schemes proposed, however, it wasn't clear whether these are sufficient conditions. Would another similar scheme give similar signatures?

**Essential References Not Discussed:**

N/A

**Experimental Designs Or Analyses:**

There seems to be extensive evaluation of the probing and patching experiments, with more results seen in the Appendix. This also enable readers to understand how early on in training the network seems to converge to these state tracking method.

A linear probe is trained, and it would be good to perform a sensitivity analysis to see how varying this probe would affect the results.

**Methods And Evaluation Criteria:**

For interpretation of how the language models are able to track state, understanding how important weights/inputs to each layer is, is important. The probing and the patching experiment seems useful to test the sensitivity of the network outputs.

**Other Comments Or Suggestions:**

There is a lot going on in Figure 1, some of which I find difficult to interpret. For example, for the probing signature, what is represented by the two bifurcating dashed lines for state parity probe for the sequential and AA algorithm?

Minor points:

Line 12: jylin04 reference seems to have incorrect author name.

Line 151: $h_{lt}$ -> $h_{t,l}$

Line 291: lenght -> length

There is a mix of using $\S$ and Section to refer to section, which can be unified.

**Other Strengths And Weaknesses:**

Overall this is a relatively well-written paper, which explores the mechanistic understanding of language models and has novel results.

Some explanations can be make clearer (see below).

**Questions For Authors:**

How sensitive is activation patching across different data samples $x$ or alternative $x'$?

**Relation To Broader Scientific Literature:**

This paper sits well in the literature of mechanistic understanding and comprehending how transformers may achieve their state tracking abilities. It's interesting to see how one of the mechanisms learned is similar to the associative scan.

**Theoretical Claims:**

There are no theoretical claims in this paper.

---

> ### Author Rebuttal · Authors · 2025-04-01
>
> We appreciate your thoughtful review, especially the time you took to examine the full submission, including the Appendix. We’re also grateful for your affirmation of our work’s contribution to the literature. Below, we address some of your specific questions:
>
> ## Unclear if signatures are sufficient conditions. Would an alternate scheme give similar signatures?
>
> You raise an important point: the signatures we describe are necessary, but not sufficient, conditions for the proposed algorithms. We tried to make a distinction between theoretical “algorithms” and empirical “mechanisms” in the paper, but to make this distinction clearer, we will add the following clarification at the beginning of Section 4 in the final version:
>
> > It is important to emphasize that the various signatures described above provide necessary, but not sufficient, conditions for implementation of the associated algorithm; the exact mechanism that LMs use in practice is likely complex and dependent on other input features not captured by the algorithms described above.
>
> ## Sensitivity analysis for linear probe: how would varying the probe affect results
>
> We trained the probe across 10 different random data subsets on the S3 AA and PAA models, and measured the standard deviations across these runs.
>
> | Layer | S3 Pythia-160M (PAA) State Std | S3 Pythia-160M (PAA) Parity Std | Pythia-160M (AA) State Std | Pythia-160M (AA) Parity Std |
> |-------|-------------------------------|----------------------------------|----------------------------|-----------------------------|
> | 0 	| 4.16e-06                  	| 3.03e-06                     	| 6.33e-06               	| 2.71e-06                	|
> | 1 	| 3.36e-06                  	| 3.31e-06                     	| 2.62e-06               	| 2.82e-06                	|
> | 2 	| 4.52e-06                  	| 5.45e-06                     	| 1.46e-06               	| 8.97e-07                	|
> | 3 	| 4.31e-05                  	| 5.59e-04                     	| 5.00e-06               	| 1.38e-06                	|
> | 4 	| 1.58e-05                  	| 3.72e-05                     	| 4.16e-06               	| 1.11e-06                	|
> | 5 	| 1.70e-05                  	| 5.38e-06                     	| 1.97e-06               	| 1.11e-06                	|
> | 6 	| 1.87e-05                  	| 7.02e-06                     	| 4.73e-06               	| 1.59e-06                	|
> | 7 	| 1.29e-05                  	| 1.34e-05                     	| 7.79e-06               	| 3.05e-06                	|
> | 8 	| 3.18e-05                  	| 3.08e-05                     	| 1.18e-05               	| 2.64e-06                	|
> | 9 	| 2.96e-04                  	| 6.85e-05                     	| 2.36e-05               	| 3.53e-06                	|
> | 10	| 3.10e-04                  	| 7.98e-05                     	| 2.60e-05               	| 2.96e-06                	|
> | 11	| 1.86e-04                  	| 8.22e-05                     	| 8.83e-06               	| 2.77e-06                	|
> | 12	| 7.05e-05                  	| 4.01e-05                     	| 1.20e-06               	| 2.75e-06                	|
>
>
> The standard deviations are all quite small, indicating that our results are highly insensitive to randomness in training.
>
> Please let us know if this aligns with what you had in mind for “sensitivity analysis,” or something else!
>
> ## Sensitivity of activation patching to varying $x$ or $x’$
>
> We performed activation patching experiments across 200 different input pairs $(x,x’)$. We plot a heatmap of the standard deviations across various (layer, position) of these pairs in:
>
> * S3 AA: https://ibb.co/5gmqB2Q2
> * S3 PAA: https://ibb.co/R4YGwC4z
>
> Note that for the PAA models, the standard deviations are highest in the middle chunk — for half the prompt pairs (when parity is equal), replacing the middle chunk results in the corrected answer, while for the other half, replacing the middle chunk has little to no effect.
>
> ## In Figure 1, what is represented by the two bifurcating dashed lines for state parity probe?
>
> The dashed lines refer to the interesting fact that for models implementing AA, some encode parity linearly, and some do not. We go into more detail on this in Appendix C.1. We will add the following to the caption of Figure 1 to avoid confusion:
>
> “Note the dotted lines indicate two different probing signatures that would both be consistent with this algorithm (see Appendix C.1 for more details).”
>
>
> ## jylin04 seems incorrect
>
> We were referencing a blog post [1] and weren’t sure about the citing format ourselves, but we think the reference is correct given that we don’t know the authors’ full names.
>
> ## Other typos / inconsistencies
>
> We will correct, thanks!
>
> [1]  OthelloGPT learned a bag of heuristics  https://www.lesswrong.com/posts/gcpNuEZnxAPayaKBY/othellogpt-learned-a-bag-of-heuristics-1

---

> > ### Comment · Reviewer_jeEg · 2025-04-05
> >
> > Thank you for your response.
> >
> > The sensitivity analysis for the probe seem good.
> >
> > For jylin04, I think looking by the other posts of this user that they may be Jennifer Lin (https://www.lesswrong.com/posts/nRu92PXLrdwqdtQmn/more-recent-progress-in-the-theory-of-neural-networks-1), but agreed can keep it as jylin04 (it just looked slightly odd on the page upon first reading).
> >
> > The suggested additions to the texts regarding sufficiency and pointing to Appendix C are useful thanks!
> >
> > For the sensitivity analysis on $x$ and $x'$, is it possible to see the PAA plot split into the two cases discussed?

---

> > > ### Author Response · Authors · 2025-04-07
> > >
> > > Yes, here is the standard deviations heatmap for PAA split into the
> > > * equal parity case: https://ibb.co/6cxM9MR4
> > > * different parity case: https://ibb.co/wq8kFBJ
> > >
> > > Note the relatively small magnitudes of the standard deviations in each case. The equal parity stddevs resembles those of the AA algorithm (as parity complement is computed associatively), while the different parity one appears to have nonzero stddevs throughout the middle chunk, depending on at what point parity gets computed.

---

### Decision · Program_Chairs · 2025-05-01

**Decision:**

Accept (poster)

**Comment:**

## Summary
The paper studies how transformer language models can keep track of hidden state. It uses permutation word problems—objects moved by a sequence of swaps—as a controlled test case. Models trained or fine‑tuned on this task learn one of two procedures. The first is an associative scan that matches prior theoretical work by Liu et al. (2023) and Merrill et al. (2024). The second first computes permutation parity to narrow the output space, then refines the result with an associative scan. These two procedures differ in robustness, and the authors show that intermediate training tasks can push a model toward either one. The findings show that transformer language models can learn efficient, interpretable, and broadly applicable state‑tracking routines.

## Decision

This paper investigates the mechanisms by which transformers perform state tracking, proposing four possible schemes and clearly delineating expected experimental signatures. The reviewers appreciated the paper's novel and compelling experimental results, particularly highlighting the identification of an intriguing algorithm akin to associative scan and permutation parity computation. The permutation composition framework is noted for its theoretical significance and practical generalizability, serving as an insightful testbed for further studies on state tracking in transformers. Additionally, reviewers praised the clarity and quality of writing, as well as the valuable insights regarding architectural choices and interventions that influence algorithmic preferences. Overall, the paper is considered a strong contribution to the mechanistic understanding literature, concretely demonstrating state-tracking capabilities in language models. The reviewers unanimously agreed to accept this paper. I believe this paper makes a meaningful contribution to the field. Thus, I recommend this paper for acceptance at ICML.